# KRAS drives immune evasion in a genetic model of pancreatic cancer

Irene Ischenko[1], Stephen D'Amico[1], Manisha Rao[2], Jinyu Li[2], Michael J. Hayman[1], Scott Powers [2], Oleksi Petrenko [1,3✉] & Nancy C. Reich [1,3✉]

Immune evasion is a hallmark of KRAS-driven cancers, but the underlying causes remain unresolved. Here, we use a mouse model of pancreatic ductal adenocarcinoma to inactivate KRAS by CRISPR-mediated genome editing. We demonstrate that at an advanced tumor stage, dependence on KRAS for tumor growth is reduced and is manifested in the suppression of antitumor immunity. KRAS-deficient cells retain the ability to form tumors in immunodeficient mice. However, they fail to evade the host immune system in syngeneic wild-type mice, triggering strong antitumor response. We uncover changes both in tumor cells and host immune cells attributable to oncogenic KRAS expression. We identify BRAF and MYC as key mediators of KRAS-driven tumor immune suppression and show that loss of BRAF effectively blocks tumor growth in mice. Applying our results to human PDAC we show that lowering KRAS activity is likewise associated with a more vigorous immune environment.

[1] Department of Molecular Genetics and Microbiology, Stony Brook University, Stony Brook, NY, USA. [2] Department of Pathology, Stony Brook University, Stony Brook, NY, USA. [3] These authors jointly supervised this work: Oleksi Petrenko, Nancy C. Reich. ✉email: alexei.petrenko@stonybrook.edu; nancy.reich@stonybrook.edu

KRAS is frequently associated with some of the deadliest forms of cancer. The prevailing tenet is that activating KRAS mutations underpin both establishment and maintenance of the transformed state, and therefore they are logical drug targets. Genetically engineered mouse models of KRAS mutant cancer have confirmed that tumor regression can be achieved via KRAS extinction[1–4]. The results have supported the view that inactivation of mutant KRAS is critically important for successful cancer treatment, in accordance with the oncogene addiction concept[5,6]. Significant efforts have centered on development of drugs that target RAS itself or its downstream signaling pathways, with the expectation of killing cancer cells while sparing normal cells. However, these targeted approaches have not been as successful as was hoped as today they benefit only a small minority of cancer patients (www.cancer.gov).

The past failures in developing anti-RAS therapies have been attributed to the difficulty of targeting RAS directly and to both intrinsic and acquired resistance mechanisms. The discovery of direct KRAS$^{G12C}$ inhibitors highlights the challenges of this therapeutic strategy and potential need for combinatorial strategies[7–9]. The key question from the perspective of cancer treatment is the extent to which KRAS mutant cancers retain dependence on KRAS. Although inhibition of KRAS expression in mice causes tumor regression, tumors relapse and become KRAS-independent[4,10–12]. Human and mouse mutant KRAS cell lines have been identified whose growth and tumorigenicity do not depend on oncogenic KRAS[13–15]. However, no clear biomarkers of escape pathways currently exist[8,9]. These findings call into question the degree to which cancers depend on continuous KRAS activity. They lend support to the concept that the initiation of oncogenic transformation and maintenance of the transformed state are separable, and that KRAS dependency is not a fundamental trait of KRAS-induced tumors[16–18]. While these studies support the initiating role of KRAS in cancer development, they underscore the need for a comprehensive view of stage-specific and cell type-specific cancer dependencies and novel rationale-based therapies.

In this work, we have addressed these questions by using a mouse model of KRAS-driven pancreatic ductal adenocarcinoma (PDAC)[19]. PDAC is a highly aggressive malignancy characterized by rapid progression, exceptional resistance to all forms of anticancer treatment, and a high propensity for metastatic spread. A striking feature of pancreatic cancer is that activating KRAS mutations are found in ~90% of cases. Mutations in other presumptive and validated driver oncogenes are remarkably rare[20]. Our objective was to confirm that PDAC cells surviving genetic ablation of KRAS retain their tumorigenic capacity, to identify stages of tumor progression when KRAS is essential, and to reveal signaling nodes in the KRAS pathway that are responsible for tumor maintenance. To accomplish these goals, here we use CRISPR-mediated gene editing to inactivate mutant KRAS in PDAC-derived cell lines. We show that KRAS-ablated cancer cells retain substantial tumorigenic capacity; however, they fail to evade the host immune system, triggering strong antitumor effects. Single-cell RNA sequencing of tumors reveals that KRAS ablation causes changes both in tumor cells and host immune cells. Our data indicate that the ability of mutant KRAS to modulate tumor immunity appears to be an essential component of its oncogenicity. The data imply that treatment of PDAC and, by extension, of other KRAS mutant cancers will require inhibition of KRAS and concurrent activation of immune pathways suppressed by cancer.

## Results

### Loss of KRAS reduces, but does not abolish, the tumorigenic capacity of PDAC cells. We previously described the isolation

and characterization of KRAS$^{G12D}$ p53KO mouse cell lines (termed KC) representing different stages of pancreatic cancer progression[19]. These cells have stable tumorigenic phenotypes and were chosen to model pharmacological inhibition of KRAS (Supplementary Fig. 1a, b). To that end, we eliminated oncogenic KRAS$^{G12D}$ by CRISPR/Cas9-mediated genome editing in clonal precancerous cell lines and their cancer-derived derivatives. We used sgRNA targeting KRAS which has been validated to have no off-target activity[14,15]. The effect of Kras gene editing was evaluated by Western blotting (Fig. 1a). Sequencing analysis of independent clones revealed deletions in the mutant Kras gene locus leading to a premature stop codon or an unstable and virtually undetectable KRAS protein (Supplementary Fig. 1c, d). The majority of precancerous KC cells treated with Kras sgRNA differentiated into non-proliferative colonies based on changes in cell morphology and proliferative rate (Fig. 1b). In contrast, cell lines established from the resected tumors formed viable colonies with higher frequencies, as determined from the analysis of 150 randomly picked clones (Fig. 1b). The increase in viability of KRAS-ablated cancer cells relative to precancerous cells was therefore considered to be due to various degrees of KRAS dependence for survival. Using these data, we selected four KRAS intact and four KRAS KO KC cell lines for molecular and functional studies (Supplementary Fig. 1e). A similar approach was used to inactivate endogenous Kras expression in KRAS$^{G12D}$ p53$^{R172H}$ (KPC) PDAC cell lines[21] (Supplementary Fig. 1b, e).

The KRAS KO clones showed reduced proliferation and colony-forming ability compared with parental KRAS intact cells when grown in serum-free epithelial cell medium. However, the growth rate was increased in serum-containing culture, supporting the role of oncogenic KRAS in growth factor-independence (Supplementary Fig. 2a). Likewise, KRAS knockout had no detrimental effect on cell viability in 3D non-adherent conditions (Supplementary Fig. 2a). To determine whether KRAS-ablated cells could form tumors in vivo, we implanted them into nude mice. When injected subcutaneously or into the pancreas, both KRAS intact and KRAS KO cells formed tumors, although KRAS KO tumors grew more slowly than those from KRAS intact cells (Fig. 1c and Supplementary Fig. 2b). When injected into the tail vein, both KRAS intact and KRAS KO clones formed lung and lymph node metastases. We observed that KRAS KO cells displayed reduced capacity for lung colonization but unabated capacity for lymph node metastases, indicating aggressive behavior (Supplementary Fig. 2c). Using limiting dilution assays in nude mice, we estimated that the frequency of tumor-initiating cells (TIC) ranged from 0.7% in KRAS intact KC/KPC cells to ~0.35% in KRAS KO cells (Fig. 1d). The cell lines derived from KRAS KO tumors exhibited stable loss of KRAS$^{G12D}$ expression, thus demonstrating that the malignant phenotype of KRAS-ablated cells is also stable (Supplementary Fig. 2d).

The morphology of tumors formed by KRAS intact KC/KPC cells resembled moderately differentiated adenocarcinomas, whereas loss of KRAS resulted in poorly differentiated neoplasms (Fig. 1e). The predominant sarcomatous elements in KRAS KO tumors maintained expression of pancreatic ductal markers, such as KRT19 and SOX9, but expression of mesenchymal genes such as ACTA2 (smooth muscle actin) was increased (Fig. 1e and Supplementary Fig. 2e). We used reverse-phase protein arrays (RPPA) of multiple clones to validate these findings. We calculated pathway scores from the expression data of ~300 cancer-associated proteins[22], and identified two pathways whose scores were significantly altered in KRAS knockouts: EMT and DNA damage response (Supplementary Fig. 2f). In contrast, there was no effect of KRAS loss on MAPK/ERK pathway activation, corroborating previously published data on KRAS knockouts[15,23]. In situ analysis showed that phosphorylation of ERK is increased

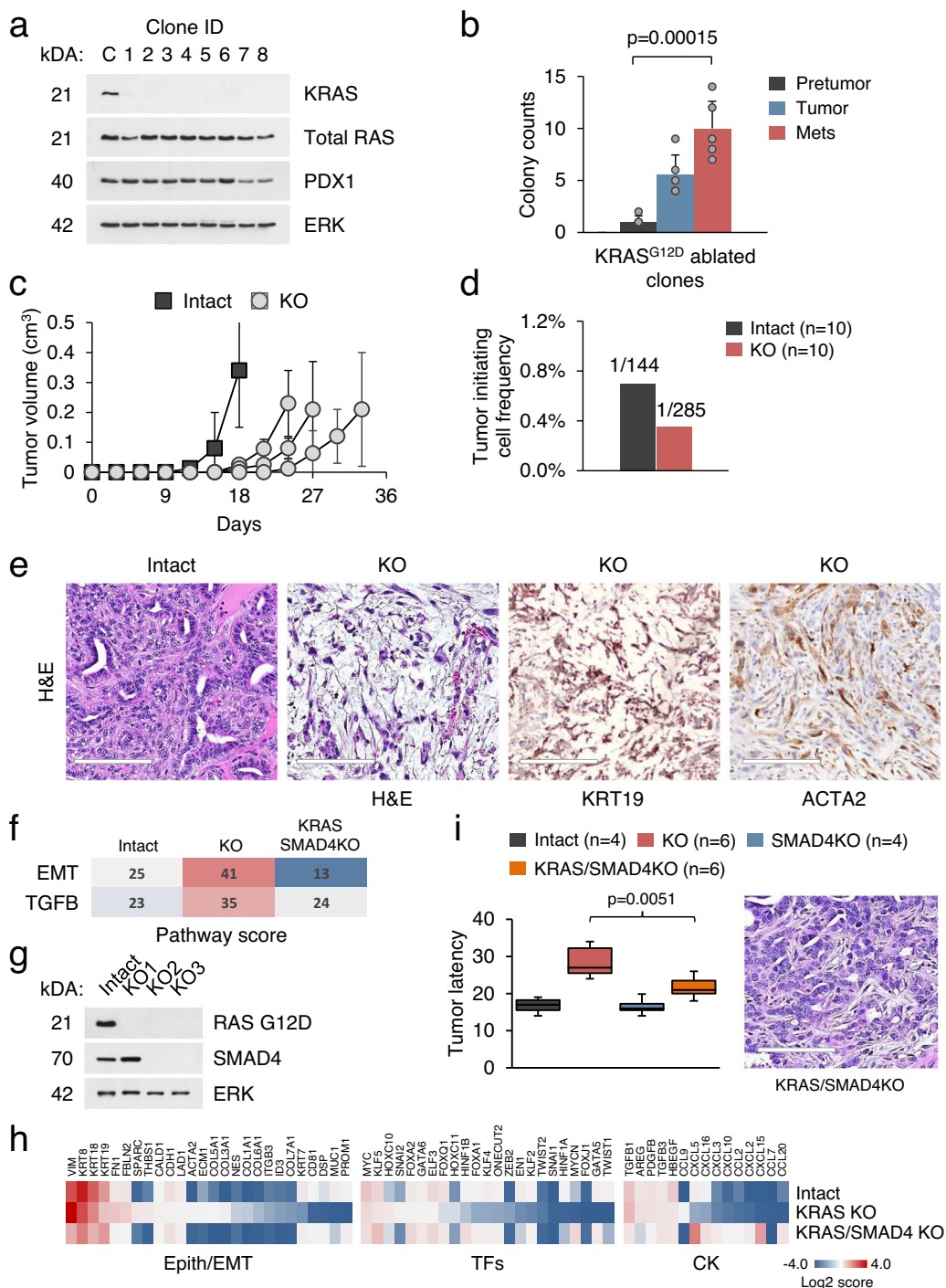

at the invasive fronts of tumors regardless of their KRAS status (Supplementary Fig. 2g). While other studies have reported induction of ERK phosphorylation by YAP1 specifically in suspended cells[10,12,24], we did not observe any alterations in YAP1 protein levels in KRAS KO cells (Supplementary Fig. 2d). CRISPR-mediated knockout of YAP1 in KRAS KO cells showed no effect on ERK activation[15].

We performed RNA sequencing of cell lines of each KRAS genotype. Data analysis revealed >700 genes that were significantly up- or downregulated (*p* values <0.05) in KRAS intact vs. KRAS KO cells. Minor variations were found in the expression of canonical RAS pathway genes or TCGA cancer driver genes[25,26], suggesting the absence of cancer-causing mutations (Supplementary Fig. 2h). In contrast, KRAS KO cells were enriched in

signatures of EMT and TGF-beta signaling (Fig. 1f). The apparent antagonism between mutant KRAS and induction of EMT was of particular interest, considering that KRAS mutant cancers are either dependent on KRAS or gain various degrees of KRAS independence through TGF-beta signaling[13,27,28]. To support this conjecture, we used CRISPR to eliminate SMAD4 in KRAS KO cell lines (Fig. 1g). Loss of SMAD4 had no effect on the proliferative properties of cells, as previously reported[27]. However, KRAS/SMAD4 KO KPC cells displayed sharply reduced expression of EMT-related genes (e.g., *Acta2, Col3a1*, and *Vim*), while RAS signature genes defined for the ability to sustain epithelial differentiation and viability of KRAS mutant cancer cells (e.g., *Cdh1, Itgb6*, and *Prom1*)[13] were among the most upregulated (Fig. 1h). Several cytokine genes were also

**Fig. 1 Loss of KRAS reduces, but does not abolish, the tumorigenic capacity of PDAC cells. a** Immunoblot analysis with anti-KRAS antibody of single-cell clones isolated following CRISPR-mediated KRAS ablation in KC cells compared to parental cells (C). For convenience, cell lines are numbered sequentially. Total RAS expression is shown. PDX1 is a specific marker for PDAC. ERK is a loading control. **b** Relative viability of KRAS KO clones derived from pretumor, tumor, and metastasis-derived KC cells ($n = 150$ individual clones examined in over three independent experiments). The cell lines are listed in Supplemental Fig. 1b. Data are expressed as mean ± SD. Significance was determined using two-tailed test at the 0.05 confidence interval. Source data are provided as a Source data file. **c** Tumor development by KRAS intact and KRAS KO KC cells transplanted subcutaneously into nude mice ($10^4$ cells per injection, $n = 20$ independent experiments). Data are presented as mean ± SD, two-tailed $t$-test. Source data are provided as a Source data file. **d** Quantification of tumor-initiating cells (TIC) in KRAS intact and deficient KPC cells. **e** Histological appearance and IHC analysis of subcutaneous tumors derived from KRAS intact and deficient KPC cells. Scale bar 200 μm. **f, g** Heatmap depicting differentially expressed pathways (**f**) and western blotting of KRAS intact KPC cells and their derivatives lacking *Kras* and *Smad4* (**g**). KRAS KO (KO1) and KRAS/SMAD4 KO (KO2 and KO3) KPC cells are shown. **h** Heatmaps of differentially expressed genes in KRAS intact, KRAS KO, and KRAS/SMAD4 KO KPC cells. Expression levels of the differentiation genes, transcription factors, growth factors, and cytokines (CK) are shown. **i** Tumor latency plot of KRAS intact KPC cells and their derivatives lacking *Kras* and *Smad4* (subcutaneous injections). Box plots show center line as median, box limits as upper and lower quartiles, and whiskers represent 1.5× interquartile range (IQR). Significance was determined using two-tailed test at the 0.05 confidence interval. Histological appearance of KRAS/SMAD4 KO tumors is shown. Scale bar 200 μm.

upregulated in SMAD4 KO cells, and *Cxcl5* and *Cxcl15* stood out as the most dramatically affected (Fig. 1h). Most notably, inactivation of SMAD4 accelerated KRAS KO tumor development in mice in a statistically significant manner and fully restored the epithelial phenotype of KRAS KO tumors (Fig. 1i). These results indicate that the loss of KRAS in pancreatic tumor cells is associated with the activation of TGF-beta/SMAD4 pathway, delay in tumor growth, and a less differentiated tumor phenotype. Inactivation of SMAD4 compensates for most, if not all, tumorigenic defects of KRAS knockout in nude mice.

**KRAS KO tumors fail to evade host immune system.** These results prompted us to reassess the role of KRAS in tumor maintenance and to explore the possibility that dependence on KRAS for tumor growth is manifested in the suppression of tumor immunity. To assess this possibility, we established a syngeneic wild-type mouse model. To that end, $5 \times 10^4$ KRAS intact and KRAS KO KPC cells were transplanted into the pancreas of syngeneic C57BL/6 mice. Cells expressing an empty CRISPR/Cas9 vector were used as controls to account for potential immune responses triggered by *Cas9* expression. Mice injected with KRAS intact cells rapidly developed tumors of differentiated PDAC and had a median survival of two weeks (Fig. 2a, b). In sharp contrast, growth of KRAS KO tumors was strongly inhibited in syngeneic mice, up to a complete rejection. Specifically, transplants with KRAS KO cells resulted in poorly differentiated tumors with latencies longer than 8 weeks (Fig. 2b, c). Moreover, >80% of mice transplanted with KRAS KO cells failed to develop tumors after 3 months of observation (Fig. 2b), suggesting that these cells are unable to evade the immune system. However, even with the low frequency of tumor formation, about 50% of KRAS KO tumors still developed peritoneal, liver, or splenic metastases (Fig. 2d and Supplementary Fig. 3a, b). Thus, metastatic pancreatic cancer is not strictly dependent on the continued expression of oncogenic KRAS. In line with this, human PDAC metastases exhibit minimal response to knockdown of oncogenic KRAS in tumor-forming assays[29]. At the same time, transplantation of merely 200–500 KRAS intact cells resulted in the formation of primary and metastatic tumors with a median animal survival of three weeks, whereas none of the KRAS KO cell lines were capable of developing tumors under these conditions after 3 months. The estimated TIC frequency of KRAS KO cells shifted from 1 in ~300 in nude mice to 1 in 120,000 in wild-type mice (Fig. 2e), while the metastatic capacity of cells shifted from 1 in ~8000 in nude mice to 1 in 230,000 in wild-type mice (Supplementary Fig. 3a, b). To further validate the importance of intact immunity, we transplanted KRAS KO KPC cells into C57BL/6 mice depleted for CD4 or CD8 T cells.

Irrespective of which T-cell population was eliminated, tumor protection was lost as these mice developed KRAS KO tumors in a timeframe similar to that in nude mice (Supplementary Fig. 3c). Overall, the results indicate that the immune system suppresses tumor-forming capacity of KRAS KO cell lines by several hundredfold (>300) compared to that in nude mice (see Fig. 1e).

We used immunohistochemistry (IHC) and flow cytometry of the resected tumors to characterize their immune composition. A pattern of significant correlation emerged, as KRAS KO tumors displayed increased lymphocyte infiltration (defined as CD45+ cells), along with areas of massive necrosis, which rarely occurred in their KRAS intact counterparts (Fig. 2f). Specifically, KRAS KO tumors displayed a high number of cells that stained positive for CD3, CD4, CD8, and CD45R, indicating both T-cell and B cell recruitment (Fig. 2g and Supplementary Fig. 3d, e). Conversely, KRAS intact tumors virtually lacked infiltrating B and T lymphocytes, while the numbers of macrophages and monocytes were not significantly different, as assessed by staining with CD11B, CD68, and F4/80 antibodies (Fig. 2g).

As a proof of concept, we performed assessment of tumor-immune interactions in mice with doxycycline-inducible KRAS[G12D/WT] pancreatic allografts[2]. In this model, KRAS[G12D] is only expressed in the presence of doxycycline, while doxycycline withdrawal renders the KRAS transgene extinct within 24 h (Supplementary Fig. 4a). Infiltrating immune cells were analyzed by IHC before and after drug withdrawal. As expected, doxycycline withdrawal initiated tumor regression, and increased the number of tumor-infiltrating T cells. The changes in tumor T cells occurred prior to tumor regression, supporting the notion that extinction of KRAS[G12D] expression triggers immune activation (Supplementary Fig. 4b, c). In contrast, we found no evidence of tumor regression in nude mouse xenografts (Supplementary Fig. 4b). Of note, tumor metastases regressed to barely detectable or undetectable levels within 2 weeks after doxycycline withdrawal (Supplementary Fig. 4d, e). To determine if KRAS extinction rendered tumors susceptible to immune checkpoint inhibition, we treated mice with anti-CTLA4 and anti-PD1 antibodies. Autopsies revealed that this led to nearly complete regression of both primary tumors and metastases in mice maintained in the absence of doxycycline, but not in its presence (Supplementary Fig. 4f). In sum, oncogenic KRAS dependency is most strongly manifested in mice with an intact immune system.

**Single-cell transcriptome profiling of KRAS intact and KRAS KO tumors.** To support these observations, we performed single-cell RNA sequencing of KRAS intact and KRAS KO tumors (10 K pooled cells from two different experiments). The transcriptome

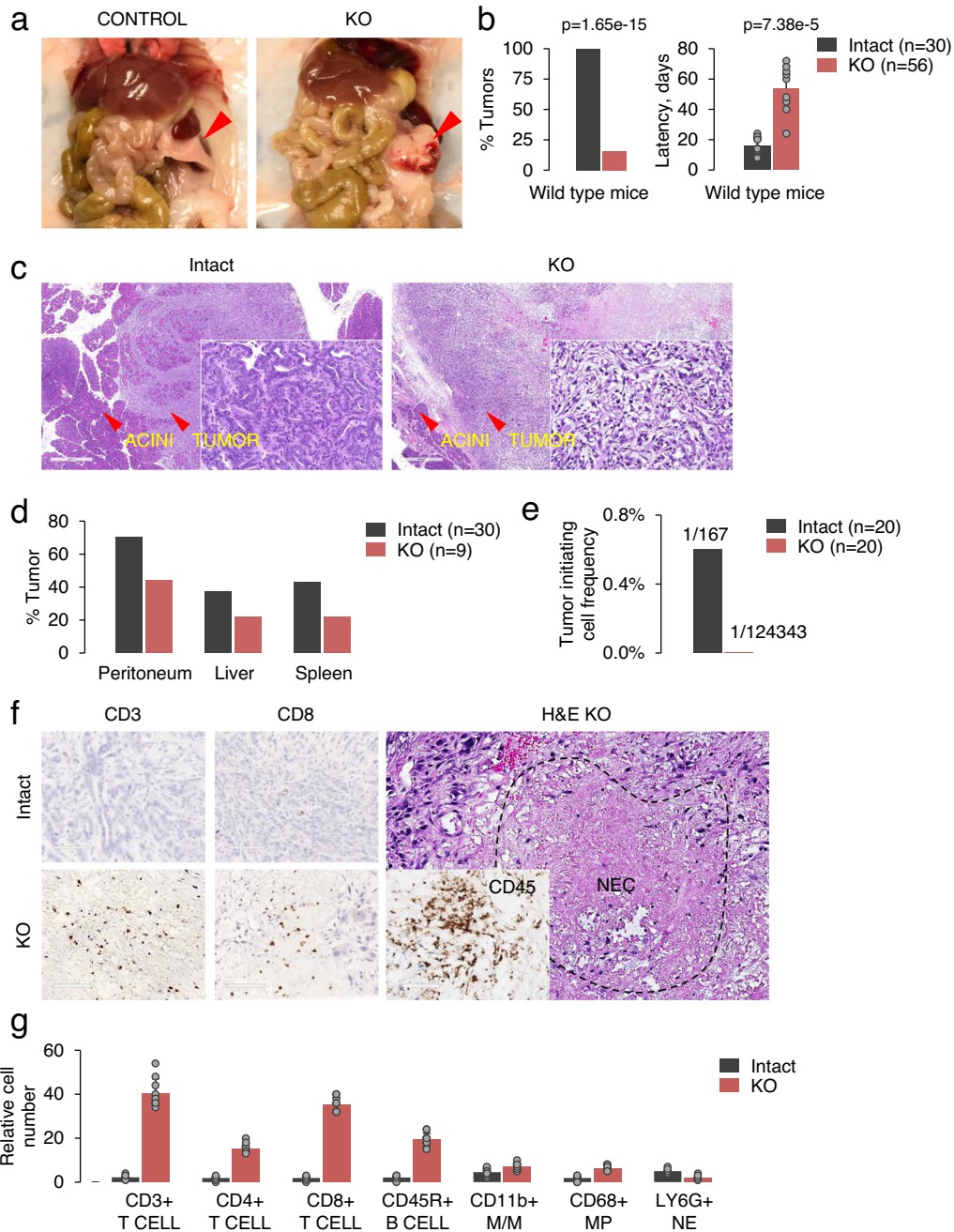

**Fig. 2 KRAS KO tumors fail to evade host immune system. a** Orthotopic pancreatic tumor in wild-type C57BL/6 mice derived from KRAS KO KPC cells. Normal pancreas is shown on left (control). **b** Bar graphs showing the quantification of wild-type mice with tumors formed by KRAS intact and KRAS KO KPC cells and the respective tumor latencies ($n = 86$ individual mice). Data are expressed as mean ± SD. Significance was determined using Fisher's exact test (left panels) or two-tailed $t$-test at the 0.05 confidence interval (right panels). **c** Histological appearance of orthotopic tumors arising from KRAS intact and KRAS KO KPC cells. Scale bar 600 μM. **d** Bar graphs display the quantification of mice with metastatic foci. **e** Quantification of tumor-initiating cells (TIC) in KRAS intact and KRAS KO KPC cells in wild-type mice. **f** IHC staining of tumors shown in (**c**) with antibodies noted. Areas of necrosis (NEC) are present in KRAS KO but not KRAS intact KPC tumors. Scale bar 100 μM. **g** Bar graph showing the quantification of tumor-infiltrating immune cells in KRAS intact and KRAS KO KPC tumors ($n = 3$ independent experiments). Relative cell number per 10 high power fields is shown. Individual points represent separate high power fields. Data are presented as mean ± SD, two-tailed $t$-test.

data showed that KRAS intact tumors segregated into distinct cell clusters based on gene expression patterns. These included tumor cells, cancer-associated fibroblasts (CAFs), myeloid cells, and endothelial cells (Fig. 3a). We identified distinct signatures of the tumor cells: classical, linked with the expression of epithelial genes (orange, Fig. 3a), and basal-like, linked with mixed expression of epithelial and mesenchymal genes (blue, Fig. 3a)[30,31]. Similarly,

CAFs fell into two groups bearing features of inflammatory fibroblasts (iCAFs) and myofibroblasts (myCAFs)[32] (Fig. 3b). Although there are different types of macrophages with varying cell surface markers, categorized as M1 (antitumor) and M2 (pro-tumor)[33], cells clustered as tumor-associated macrophages (TAMs) showed a mixed phenotype expressing both M1 and M2 markers (Supplementary Fig. 5a).

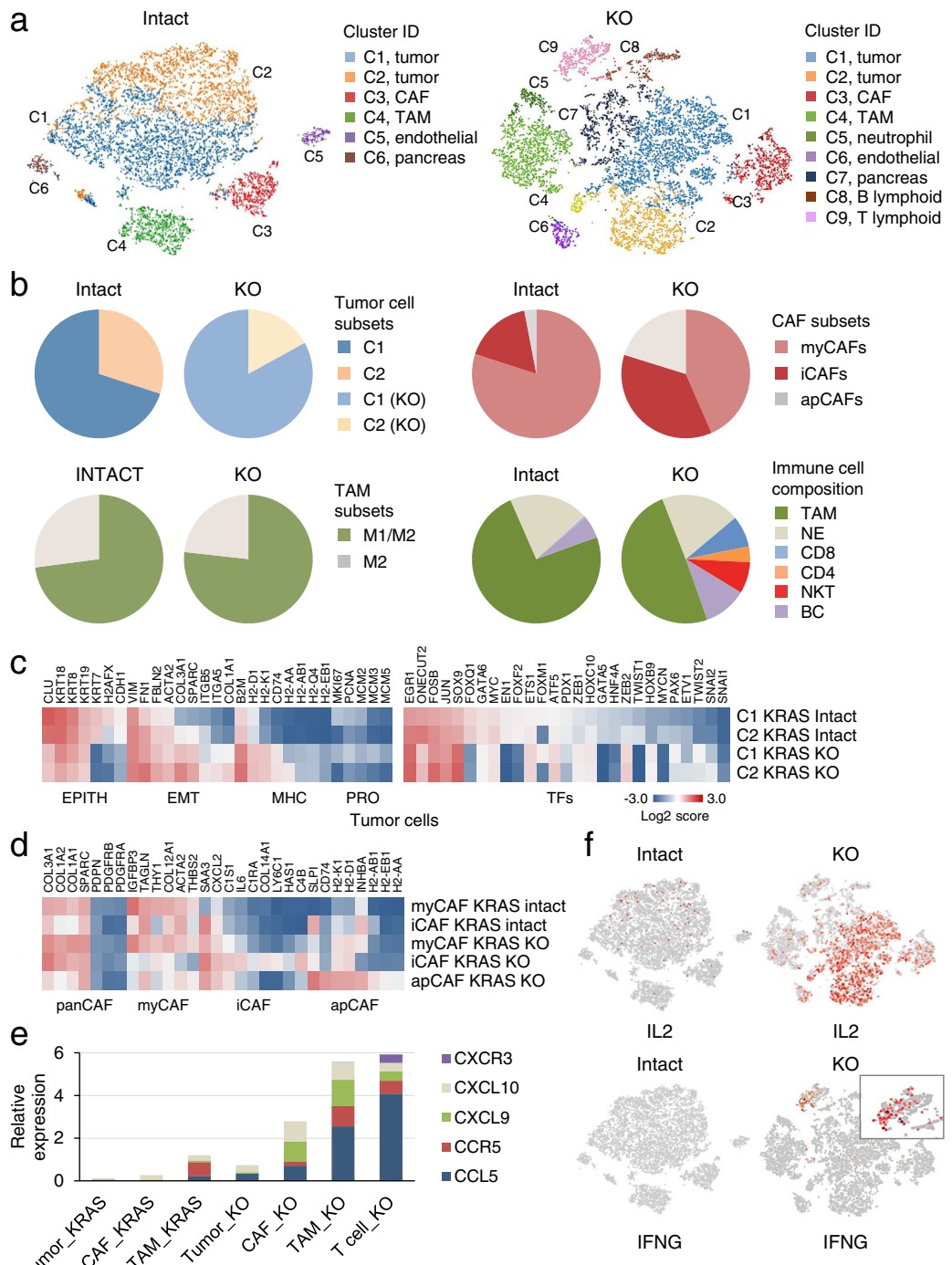

**Fig. 3 Single-cell transcriptome profiling of KRAS intact and KRAS KO tumors. a** Single-cell clustering of KRAS intact and KRAS KO KPC tumors. Individual clusters are indicated by colors. **b** Pie charts of individual cell types identified from single-cell transcriptomic data. **c, d** Heatmaps of differentially expressed genes in tumor cells (**c**) or CAFs residing in tumors (**d**). **e** Cytokine and cytokine receptor gene signatures in KRAS intact and KRAS KO KPC tumors. **f** t-SNE plots showing expression of selected genes in KRAS intact and KRAS KO KPC tumors.

Defining KRAS KO tumor cell types was more intricate than those of KRAS intact tumors owing to massive changes in gene expression (Fig. 3a). Striking phenotypic changes were apparent in both KRAS KO tumor cells and stromal cells (Fig. 3b). The tumor cells showed a shift from epithelial to a mesenchymal-like gene signature manifested by a dramatic activation of collagen genes and reduction in keratin expression (Fig. 3c). We identified transcription factors involved in the control of EMT (e.g., FOX, SOX, and ZEB gene families) that were significantly up- or downregulated (p values <0.05) in KRAS KO compared with KRAS intact cells (Fig. 3c). In contrast, there was little change in gene expression associated with proliferation (*Mki67*, *Pcna*, and *Mcm* genes) (Fig. 3c). IHC staining of KRAS KO tumors showed significant levels of ERK activation, confirming the in vitro data (see above).

We examined expression of immunomodulatory genes (IMs) in the context of TCGA clinical data[34]. A striking feature of KRAS KO tumor cells was a significant 2–10-fold increase in MHC gene expression (*H2d1*, *H2k1*, *B2m*, and *H2-aa*, *H2-ab1*, *CD74*) over KRAS intact controls (Fig. 3c). These features suggest

enhanced susceptibility to T-cell immunity. A further distinction of KRAS KO tumors was the presence of myCAFs (high *Acta2* and *Thy1* expression), iCAFs (high levels of inflammatory cytokines), and antigen-presenting CAFs (increased *CD74* and MHC class II expression)[35] (Fig. 3b). The TAMs in KRAS KO tumors were skewed toward M1 polarization (Supplementary Fig. 5a), while tumor-infiltrating lymphocytes (TILs) were dominated by CD8 cytotoxic T cells (40%) and natural killer T (NKT) cells (40%), compared to CD4 helper T cells (20%). In-depth analysis of these cell populations revealed interferon-gamma (IFNG) production, suggesting a shift to a Th1 (inflammatory) response (Supplementary Fig. 5b). Receptors with stimulatory (CD28, CD69, ICOS) and inhibitory roles (CTLA4, HAVCR2, LAG3, and PDCD1) were expressed by both CD4 and CD8 T cells, indicating abundant activity (Supplementary Fig. 5b, c).

We examined whether Th1 pathway (cellular immune response) was affected by KRAS loss. We identified differential expression of cytokines (*Ccl2*, *Ccl5*, *Ccl7*, *Cxcl2*, *Cxcl9*, and *Cxcl10*) that are known to mediate T-cell recruitment and differentiation (Supplementary Fig. 5e). Each of these cytokines was elevated in KRAS KO tumors. Other cytokines reported to be associated with tumor immunity (e.g., *Csf1-3* and *Ccl9*) were not differentially expressed[36]. Among the ligand–receptor pairs, only the expression of CCL5/CCR5, CXCL9/CXCR3, and CXCL10/CXCR3 significantly correlated with the presence of TILs (Fig. 3e). Data analysis revealed a strong association between expression of these chemotactic genes and loss of KRAS (p < 0.0005 by two-way ANOVA test). The gain of IFNG and IL2 expression upon KRAS loss can further enhance immune activation (Fig. 3f)[37,38]. In sum, while the immune response to cancer arguably lies in MHC-dependent tumor antigen presentation[39], loss of KRAS confers profound changes to both tumor cells and infiltrating cells.

**Immunomodulatory effects of oncogenic KRAS in human PDAC**. We set out to determine whether the immunomodulatory KRAS effects observed in the mouse models were relevant to human PDAC. Since the vast majority of human PDACs harbor oncogenic KRAS mutations, we used two non-mutational assessments of oncogenic KRAS activity. One measure was the putative KRAS-dependency gene expression signature derived from a panel of human KRAS mutant cancer cell lines that differed in the survival and differentiation on continued KRAS expression[13]; and the second was an expression signature derived from our in vivo single-cell RNA sequencing data (36 genes with significantly greater expression in the epithelial cells of KRAS intact compared to KRAS KO tumors, Supplementary Table 1).

Aligning the human PDAC samples according to the KRAS-dependency score allowed us to group the human tumors into those that showed strong KRAS dependency and those that were more KRAS-independent (Fig. 4a)[13]. Histological examination of these tumors, which we call KRAS-dep and KRAS-indep, showed well, moderately or poorly differentiated carcinomas (Supplementary Fig. 6a). Similarly, aligning the human samples according to their in vivo KRAS signature score segregated tumors into two groups, one with high KRAS activity and one with low activity (Supplementary Fig. 6b). We observed a significant correlation between the two KRAS oncogenic activity scores (r = 0.55), indicating that these two scores measure overlapping KRAS-induced transcriptional alterations in human tumors (Supplementary Fig. 6c). We also observed nonrandom patterns of mutations in KRAS (94% in KRAS-dep versus 60% in KRAS-indep), TP53 (74% in KRAS-dep versus 42% in KRAS-indep), CDKN2A (54% in KRAS-dep versus 17% in KRAS-indep), and SMAD4 (50% in KRAS-dep versus 15% in KRAS-indep) (Fig. 4a). Thus, KRAS dependency is strongly linked to both

activation of KRAS and inactivation of tumor suppressor genes. KRAS wild-type tumors were dispersed along the range of values for both expression scores, consistent with the requirement for strong MAPK/ERK signaling in KRAS wild-type PDAC[20]. Notably, KRAS-indep tumors showed enriched expression of mesenchymal stromal genes and reduced expression of epithelial genes (Fig. 4b).

Comparison of gene expression levels of immunity genes between KRAS-dep and KRAS-indep tumors also showed striking differences, with KRAS-indep tumors showing higher levels of genes expressed in CD4 and CD8 T cells (Fig. 4c and Supplementary Fig. 6e). Similar results were obtained when the human PDAC samples were segregated by the in vivo KRAS signature score into KRAS-high and KRAS-low groups (Supplementary Fig. 6f). These gene expression results suggesting greater immune infiltration in KRAS-indep and KRAS-low tumors were corroborated by analysis of T-cell infiltration based on TCGA digital-pathology examination of the tumor tissue slides (Fig. 4d)[40]. When grouped by the immune subtype, KRAS-dep tumors fell into four categories: C1 (termed wound healing), C2 (IFNG dominant), C3 (inflammatory), and C6 (TGF-beta dominant) (Fig. 4e). In contrast, KRAS-indep tumors were significantly skewed toward the C3 subtype (p = 0.0045, two-sided Fisher's exact test) (Fig. 4e). This subtype is defined by a high Th1/Th2 ratio, low to moderate tumor cell proliferation, and the most favorable prognosis[34]. The KRAS-low group showed a similar enrichment in the C3 immune subtype (p = 0.002, Supplementary Fig. 6g). In addition, the KRAS-indep and KRAS-low groups showed greater tumor leukocyte fraction, computed by TCGA from DNA methylation profiles[20]. Both the KRAS-indep and KRAS-low groups showed significantly higher leukocyte infiltration when compared to the KRAS-dep and KRAS-high groups, respectively (p < 0.05, Fig. 4f). Overall, TILs and TAMs were the most abundant immune cells in the tumor defined by their production of inflammatory cytokines and hence the establishment of inflammation (p < 0.05, Supplementary Fig. 6h, i). We also investigated the expression of immune checkpoints. The expression of *Ctla4*, *Pdcd1*, and *Pdl2* was found to be higher in the KRAS-low groups, making them potentially more amenable to treatment (Fig. 4g). Taken together, these results strongly support the proposition that oncogenic KRAS suppresses antitumor immunity response in mouse as well as human PDAC.

**BRAF and MYC partially rescue tumorigenesis of KRAS KO cells in immunocompetent mice**. We next sought to identify the downstream effectors of KRAS-driven immune suppression. As the first step, varying amounts of KRAS/SMAD4 KO cells were transplanted into wild-type C57BL/6 mice. However, six out of eight mice transplanted with these cells still failed to develop tumors at endpoint 10 weeks after injection. This suggests that the immunosuppressive impact of KRAS mutation extends beyond the regulation of TGF-beta/SMAD4 signaling. We therefore generated KRAS KO KPC cell lines stably expressing gain and loss of function mutants in RAS pathway components. Genes were introduced by retroviral transduction and an empty retroviral vector was used as control (Supplementary Fig. 7a). We found that a subset of genes, for instance activated *Braf* and *Akt1*, did accelerate KRAS KO tumor formation in nude mice (Fig. 5a). However, these same genes failed to functionally replace KRAS$^{G12D}$ and initiate tumor formation in wild-type mice (Fig. 5a). For example, expression of *Braf*$^{V600E}$ accelerated KRAS KO tumor development in nude mice twofold, but had no discernible effect in wild-type mice (Fig. 5a).

We therefore examined pairwise gene combinations. Among gene pairs tested, the combined expression of *Braf*$^{V600E}$ and *Myr-Akt1* partially restored KRAS KO tumor formation in wild-type

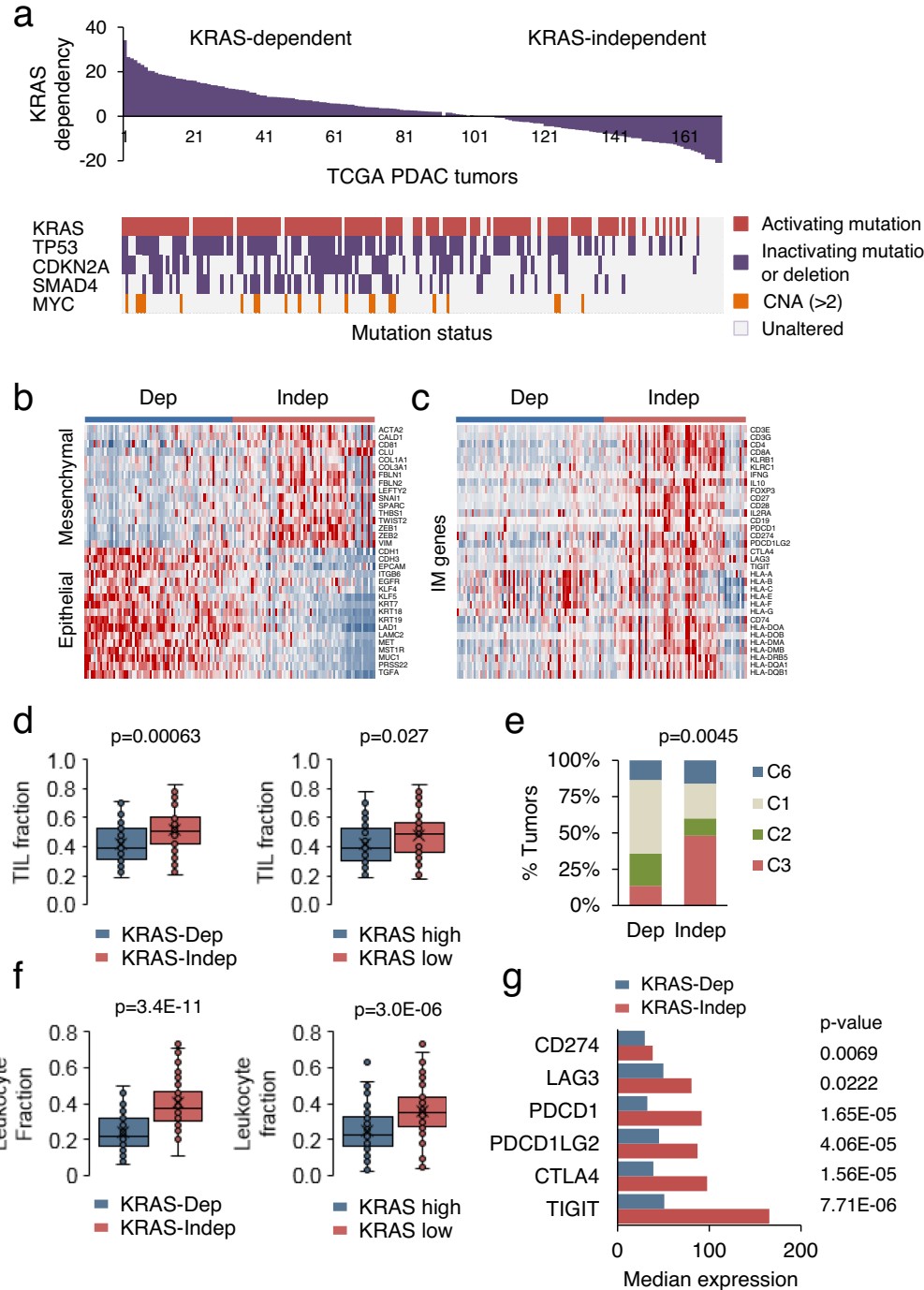

**Fig. 4 Immunomodulatory effects of mutant KRAS in human PDAC. a** Human PDAC samples (*n* = 178) derived from the TCGA database were divided into two groups according to a gene expression signature (KRAS-dependency index, RDI) that distinguishes putative KRAS-dependent (RDI > 4) and -independent cell states (RDI < 0) (top). Landscape of genomic alterations in the human PDAC tumors is shown (bottom). **b**, **c** Heatmaps derived from data in (**a**) depicting differentially expressed genes in KRAS-dependent tumors (*n* = 69) versus distinctive KRAS-independent tumors (*n* = 65). **d** Box plots showing TIL fraction in KRAS-dependent (*n* = 60) versus KRAS-independent tumors (*n* = 50), and KRAS-high (*n* = 60) versus KRAS-low tumors (*n* = 50) in the digital-pathology PDAC database[40]. Box plots show center line as median, box limits as upper and lower quartiles, and whiskers represent a 1.5× interquartile range. Significance was determined using two-tailed *t*-test at the 0.05 confidence interval. **e** Distribution of immune subtypes within KRAS-dependent (*n* = 60) and KRAS-independent tumors (*n* = 50): C1 (wound healing), C2 (IFNG dominant), C3 (inflammatory), and C6 (TGF-beta dominant). The proportion of samples belonging to each immune subtype is shown. Significance was determined using two-sided Fisher's exact test. **f** Box plots showing leukocyte fraction in KRAS-dependent (*n* = 68) versus KRAS-independent tumors (*n* = 52), and KRAS-high (*n* = 68) versus KRAS-low tumors (*n* = 52) in the immune PDAC database[34]. Box plots show center line as median, box limits as upper and lower quartiles, and whiskers represent a 1.5× interquartile range. Significance was determined using two-tailed *t*-test at the 0.05 confidence interval. **g** Differential expression of immune checkpoint genes in KRAS-dependent (*n* = 68) and KRAS-independent tumors (*n* = 52). Significance was determined using two-tailed *t*-test at the 0.05 confidence interval. Tumors from **b**, **c** for which the immune information is available are considered in panels (**d**–**g**). Source data are available as a Source data file.

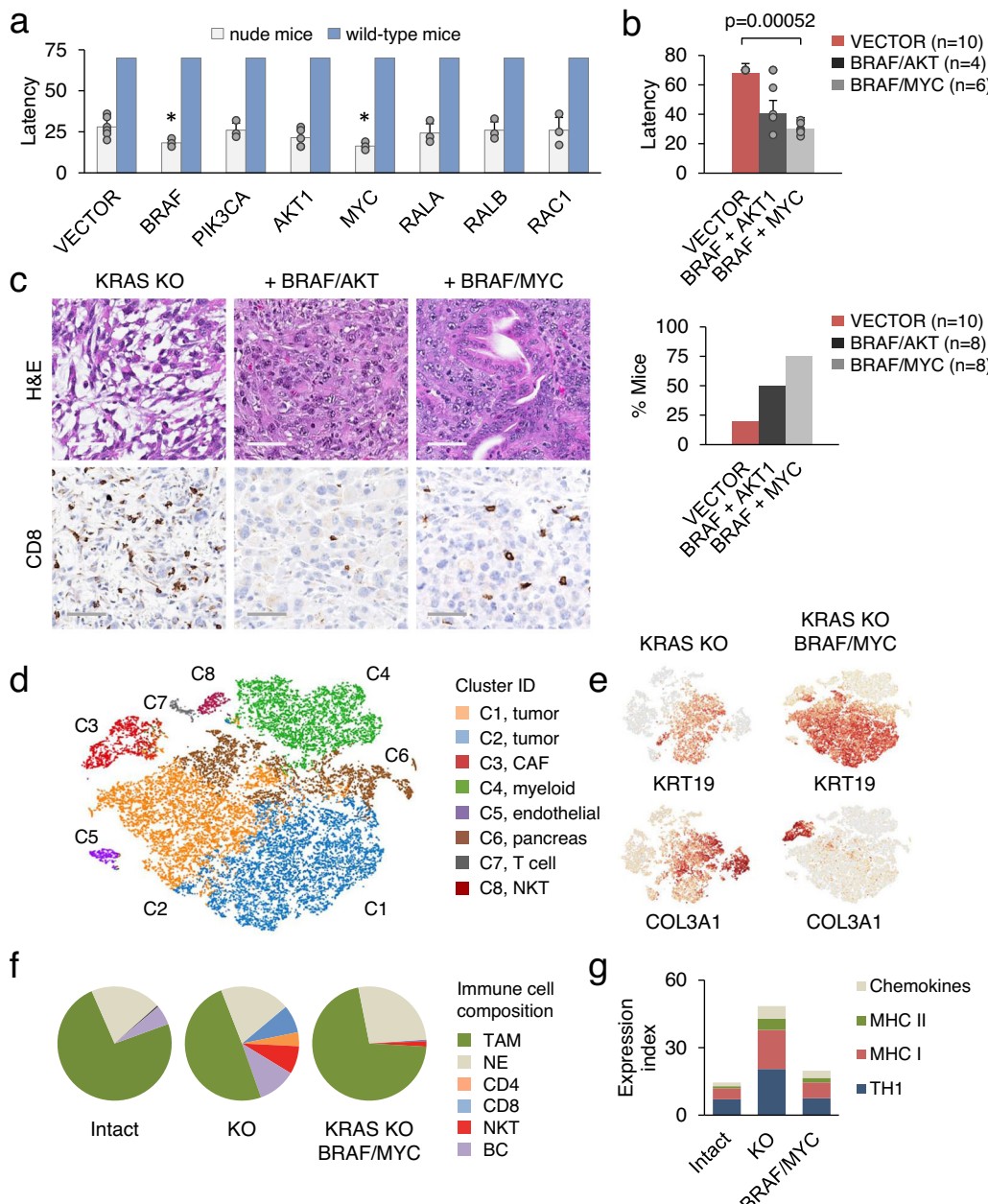

**Fig. 5 BRAF and MYC partially rescue tumorigenesis of KRAS KO cells in immunocompetent mice. a** Latency of tumor development in nude mice or wild-type mice by retrovirally transduced KRAS KO KPC cells relative to vector infected cells. Results are derived from three independent injections of each cell line. Non-tumor-bearing mice were terminated on day 70 post-implantation. Data are presented as mean ± SD. Significance was determined using two-tailed test at the 0.05 confidence interval. Asterisks indicate statistically significant differences ($p = 0.0365$ for *Braf* and 0.017 for *Myc*, respectively). **b** The mean latency (top) and efficiency (bottom) of tumor development by KRAS KO KPC cells transduced with the indicated genes. Data are expressed as mean ± SD. Significance was determined using two-tailed *t*-test at the 0.05 confidence interval. **c** Histology and IHC staining of representative tumors shown in (**a**). Scale bar 100 μM. **d** Single-cell clustering of KRAS KO tumors reconstituted with *Braf*[V600E] and *Myc*[T58A]. The data represent 24 K pooled cells from two different tumors. **e** t-SNE plots showing expression of selected genes in KRAS KO KPC tumors before and after reconstitution with *Braf*[V600E] and *Myc*[T58A]. **f** Immune cell composition of KRAS intact and KRAS KO KPC tumors before and after reconstitution with *Braf*[V600E] and *Myc*[T58A]. **g** The mRNA expression levels of Th1, MHC I, MHC II, and chemokine genes (*Ccl5*, *Cxcl9*, and *Cxcl10*) in KRAS intact and KRAS KO tumor cells before and after reconstitution with *Braf*[V600E] and *Myc*[T58A] (Fig. 5d). Source data are provided as a Source data file.

mice (Fig. 5b). Because both KRAS and AKT regulate proteins involved in the MYC network, including MYC-MAX and MAD-MAX[41,42], we tested if enhanced *Myc* expression could substitute for the loss of KRAS. Suppression of antitumor immunity by *Myc* is an important facet of its activity in cancer[43,44]. Although *Myc* alone had no effect on the formation of tumors in wild-type mice, KRAS KO cells co-expressing *Braf*[V600E] with wild-type or stabilized *Myc*[T58A] showed the capacity to generate xenografts

and to reproduce the morphological characteristics of KRAS intact tumors (Fig. 5b, c). These changes in differentiation of KRAS knockouts were accompanied by a reduction in both absolute and relative number of TILs (Fig. 5c and Supplementary Fig. 7b). In contrast, *Braf*[V600E] and *Myc* had no discernible effect on tumor cell proliferation (Supplementary Fig. 7c, d).

Single-cell RNA sequencing analysis confirmed that KRAS KO KPC tumors with *Braf*[V600E] and *Myc*[T58A] expression regained

their epithelial phenotype, presumably by reversing EMT and promoting MET (Fig. 5d, e). TAMs represented the major component of the immune infiltrate, while TILs accounted for only a small minority (~0.5%) of tumor cell population, and their lack was aggravated by the total absence of CD4 T cells (Fig. 5f and Supplementary Fig. 7e). The expression of genes associated with immunomodulation (Th1), antigen presentation (MHC I and MHC II), and T-cell activation (*Ccl2*, *Ccl5*, and *Cxcl10*) also resembled KRAS intact tumors, suggesting a common mechanism of immune evasion (Fig. 5g). Thus, activation of BRAF and MYC can partially substitute for oncogenic KRAS and convert T-cell-inflamed tumors into cold tumors.

**Inactivation of BRAF impairs the growth of KRAS mutant PDAC**. We next set out to use our isogenic KRAS KO system to elucidate the functional impact of KRAS loss on MAPK signal transduction. To that end, cells were maintained in 2D adherent or 3D suspension conditions. Comparison of KRAS intact cells grown in 2D compared to 3D cultures showed that phosphorylation of CRAF and MEK was blunted in suspended cells, while activation of ERK remained unperturbed (Fig. 6a). In clear contrast, both 2D and 3D KRAS KO cultures displayed reduced levels of CRAF and MEK phosphorylation, regardless of the culture conditions. However, basal ERK activation, as determined by phosphorylation of ERK, was marginally affected (Fig. 6a). These results are consistent with a predominantly MEK-independent mode of ERK activation in KRAS KO cells, where the RAF/MAPK axis appears to be partially silenced despite remaining structurally intact. These data support the concept that cancer cells can circumvent their requirement for RAS-dependent mechanism of ERK activation[15,23]. Likewise, loss of KRAS attenuated the activating phosphorylation of AKT, but not RSK or S6K (Fig. 6a). GSK3 phosphorylation was also diminished in KRAS KO cells, consistent with its position downstream of AKT (Fig. 6a). Indeed, as reported, KRAS is a selective RAS isoform responsible for basal and growth factor-induced AKT activation[45].

CRAF is known to be impaired in suspended cells due to the loss of S338 phosphorylation[46,47]. Moreover, CRAF was found to be dispensable for KRAS-induced pancreatic cancer[48,49]. Thus, BRAF may be preferentially required for MAPK pathway and, by extension, for PDAC initiation and/or maintenance in the context of oncogenic KRAS. To assess the role of BRAF in PDAC maintenance, we generated a panel of BRAF KO KPC cell lines with active KRAS^{G12D} expression (Fig. 6b). Optimized *Braf* sgRNAs with minimal off-target activity were used to reduce the risk of side effects[50]. We asked whether *Braf* is essential for KRAS-induced tumor growth and whether its loss affects tumor-immune interactions. We found significant similarities between *Kras* and *Braf* knockouts, including a striking disconnect between activation of RAS and MAPK/ERK pathway components (Fig. 6b). Of equal note, KPC cells lacking *Braf* expression displayed no proliferation defects in vitro and were able to form tumors in nude mice at 100% efficiency within 1 month. The calculated frequency of tumor-initiating BRAF KO cells (TIC) was only 4–5-fold lower than that of BRAF intact controls (Fig. 6c). However, these cells formed tumors in wild-type mice >500-fold less efficiently compared to wild-type controls (Fig. 6c). By examining recipient wild-type mice after two months, we found no tumors in nine out of ten animals transplanted with different numbers of BRAF KO cells (ranging from 500 to 50,000 cells per injection). Optical imaging of Luc-expressing clones and the histological examination of pancreatic tissue in each sample group confirmed these observations (Fig. 6d, e). Notably, IHC staining for CD8 T cells in the one BRAF KO tumor showed that even with this

one sample, there was significant T-cell infiltration of the tumor (Fig. 6f). Thus, pancreatic tumor growth in wild-type mice is impaired by loss of BRAF. A more nuanced understanding of how BRAF may limit the host immune response will depend on the discovery and development of adequate RAF kinase inhibitors[49].

**MYC and SMAD4 play opposing roles in pancreatic tumor maintenance**. We examined the possibility of cooperation and reciprocity between activation of the MYC pathway and inactivation of the TGF-beta pathway. We used TCGA data of PDAC to assess patterns of mutual exclusivity and co-occurrence between gains of *Myc* and inactivation of *Smad4*, as mutual exclusivity can involve genes that are synthetically lethal. The TCGA dataset revealed a strong tendency toward co-occurrence between gains of *Myc* and *Smad4* mutations ($p = 0.011$), and it was particularly evident in KRAS-dependent tumors (Supplementary Fig. 6b). Our single-cell RNA sequencing data demonstrated that the expression of TGF-beta target genes associated with cell adhesion and EMT (e.g., *Acta2*, *Col3a1*, *Snai1*, *Twist1*, *Zeb1*, etc.) was reduced in BRAF/MYC tumors to the level observed in KRAS intact controls (Fig. 7a). The same was true for MYC target genes, particularly those involved in RNA processing and protein synthesis (Fig. 7b). As stated above, one of the most obvious features of BRAF/MYC tumors was a reversal of the inflammatory gene expression (Supplementary Fig. 5d). Given the distinct pattern of these gene expression changes, we investigated whether inactivation of *Smad4* could facilitate tumorigenesis driven by *Braf* and *Myc*. Indeed, loss of *Smad4* rendered wild-type mice more susceptible to BRAF-driven tumorigenesis, while combined expression of activated *Braf* and *Myc* in the background of *Smad4* knockout nearly completely compensated for KRAS loss in promoting tumor survival and growth (Fig. 7c, d). Histologically, KRAS/SMAD4 KO tumors were moderately differentiated adenocarcinomas exhibiting glandular growth patterns (Fig. 7e). Phosphorylation of SMAD2, a surrogate marker of TGF-beta pathway activation, was not strongly affected by the loss of *Smad4*, but it was diminished in SMAD4 KO tumors with *Myc* overexpression (Fig. 7e). This was accompanied by reduced T-cell infiltration, supporting opposing roles of the SMAD and MYC pathways in shaping tumor immunogenicity (Fig. 7e, f). We surmise from these observations that MYC impairs antitumor immunity independently of its effects on cell proliferation. The clinical relevance of SMAD4 inactivation is that it may serve as an escape mechanism from oncogenic KRAS addiction in pancreatic cancer development (shown schematically in Fig. 7g).

**Discussion**

Here, we show that in an established model of KRAS-driven pancreatic cancer KRAS ablation does not affect intrinsic tumorigenic capacity, but elicits antitumor immune response. This result highlights the importance of KRAS-driven immune suppression in tumor maintenance. KRAS is the most frequently mutated oncogene in human cancer. However, effective therapies against KRAS have not yet been developed. The past failures in developing anti-KRAS therapies have been attributed to the difficulties of targeting RAS directly and to resistance based on activation of alternative routes to MAPK/ERK activation[10,12,13]. In addition, resistance to blocking KRAS can occur through receptor tyrosine kinases signaling to both PI3K/AKT and RAS/MAPK pathways[4,15,51,52]. Perhaps the best understood bypass mechanism is through rewiring of signaling networks downstream of PI3K[15,16].

The goal of this study was to determine whether pancreatic cancer cells retain their viability and tumorigenic capacity

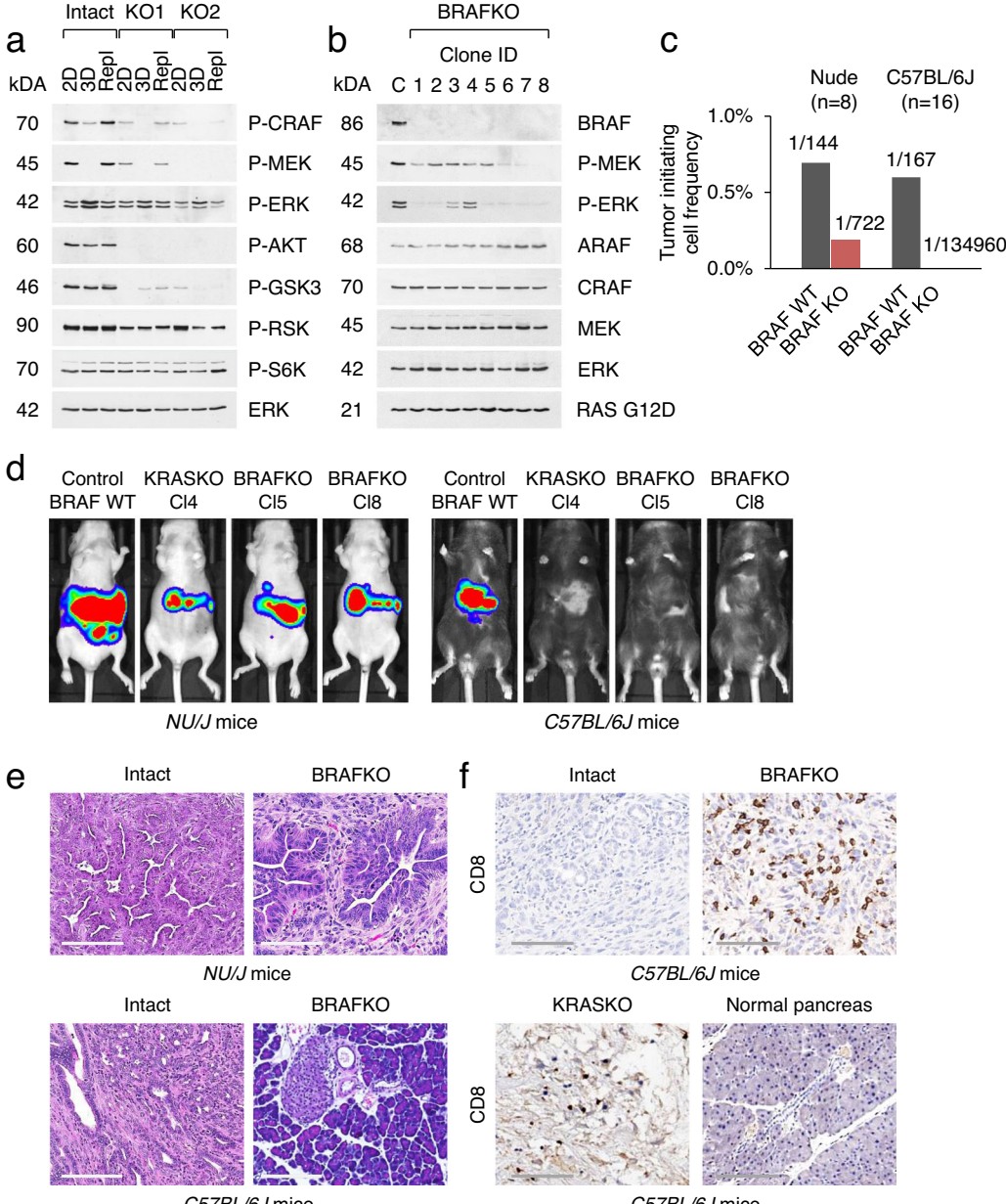

**Fig. 6 Inactivation of BRAF impairs the growth of KRAS mutant PDAC. a** Western blot analysis of parental KRAS intact and KRAS KO cell clones maintained in adherent (2D) or suspension culture (3D) for 3 days. Cells maintained in 3D culture retain adhesive capacity when replated (Repl) on tissue culture dishes. **b** Immunoblot analysis of clonal BRAF KO KPC cell lines. For convenience, cell lines are numbered sequentially. **c** Tumor-initiating cell frequency of parental KRAS intact and averaged four BRAF KO KPC cell lines (Cl5, 6, 7, and 8) in nude and syngeneic wild-type mice. Source data are provided as a Source Data file. **d** Bioluminescence imaging of tumor development by Luc-expressing cell lines of the indicated KRAS and BRAF genotypes two weeks post-implantation. Each designated cell line was transplanted into two mice, and one representative mouse is shown. **e** Histological appearance of orthotopic tumors derived from KRAS intact KPC cells and their derivatives lacking BRAF. Scale bar 200 μM. **f** IHC staining of representative tumors shown in (**d**).

independent of KRAS, to identify stages of tumor progression when KRAS is essential, and to explore changes that enable cancer cells to escape from KRAS dependence. To this end, we compared the effects of CRISPR/Cas9-mediated KRAS knockout in premalignant and cancer-derived PDAC cell lines. We quantified KRAS dependencies for these cell lines based on the accepted definition of malignant transformation: we used a defined genetic setting; we tested many independently derived KRAS knockouts; we ascertained that the phenotypes of the derived cells were stable over time; and we showed that re-introduction of KRAS expression reversed the knockout

phenotype. The results provide evidence that the malignant phenotype of KRAS knockout cancer cells is stable. The dependence on KRAS for tumor growth/survival is reduced in the majority of cell lines that we screened, and is instead manifested in the suppression of antitumor immunity. Our key observation is that KRAS depleted cells struggle to adapt to the activation of the immune system. The data imply that the anticancer immune response is indispensable for therapeutic management of KRAS-driven tumors and that combination treatment that both targets KRAS signaling and boosts antitumor immunity may be an effective strategy to treat PDAC.

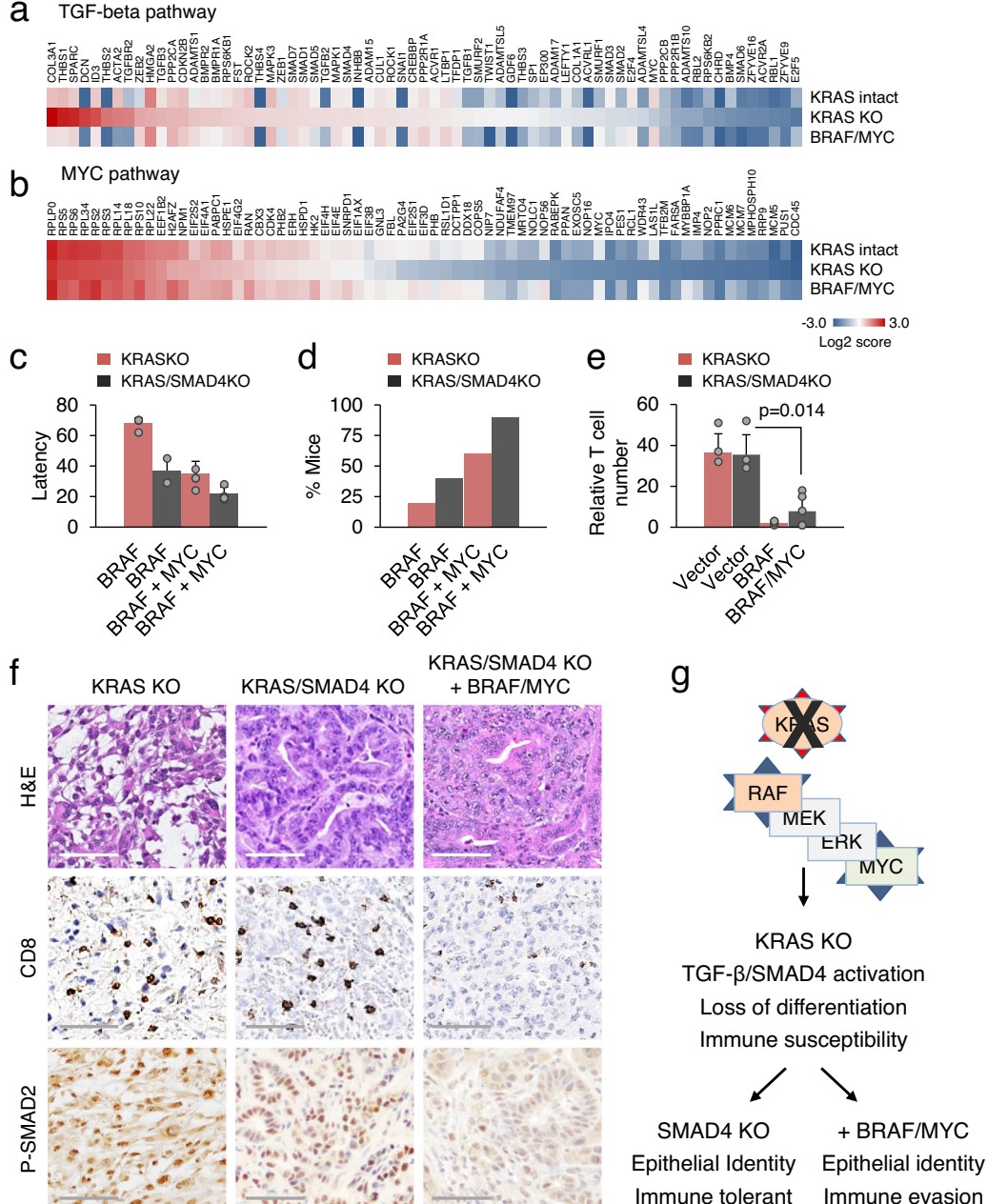

**Fig. 7 MYC and SMAD4 play opposing roles in pancreatic tumor maintenance. a, b** Heatmaps from scRNA-seq showing differentially expressed TGF-beta/SMAD and MYC pathway genes in KRAS intact, KRAS KO, and BRAF/MYC reconstituted KPC tumors. **c, d** The mean latency (**c**) and efficiency (**d**) of orthotopic tumor development in wild-type mice by KRAS KO and KRAS/SMAD4 KO KPC cells (*n* = 5 for each genotype) transduced with the indicated genes. Data are presented as mean ± SD, two-tailed *t*-test. **e, f** Histology and quantification of IHC staining of tumors (*n* = 5 for each genotype). Data are expressed as mean ± SD. Significance was determined using two-tailed test at the 0.05 confidence interval. Scale bar 200 µM. **g** Conceptual schematic of phenotypic changes in PDAC tumors. BRAF and MYC are key mediators of KRAS-induced immune evasion. Loss of SMAD4 enhances KRAS-independent tumor growth.

It remains to be determined whether the immune evasion phenotype of transplantable cancer cells can be faithfully recapitulated in the animal models of spontaneous PDAC. The current paradigm of cancer is based on the principles that cancer is a genetic disease advancing in a stepwise manner, and that to cure cancer we must target cancer-causing genes. Tumor regression following oncogene inactivation is thought to be a consequence of restoration of normal checkpoint mechanisms, cell cycle arrest, apoptosis, and differentiation of the tumor cells[5,6]. However, current models of oncogene addiction are notable for their limitations, as many KRAS mutant cell lines are not strictly dependent on continued KRAS

expression, and KRAS mutant cancers have been classified into discrete molecular subtypes based on the degree of KRAS addiction[13–15,18]. Together, these findings suggest that tumor initiation and tumor maintenance may not be as inextricably linked as previously thought. Moreover, one of the lessons learned from cancer genome sequencing is that as many as 30% of human cancers lack well-defined oncogenic driver mutations[26,53,54]. It follows that both conditions, i.e., the presence or absence of known cancer driver genes, can give rise to tumors that exhibit similar morphology and clinical characteristics, suggesting common mechanistic underpinnings. The most obvious commonality here is that in addition to

genetic factors (such as oncogenic KRAS) non-genetic factors such as stable reprogramming of cancer genomes contribute to human carcinogenesis[29,55–57]. Our observation that tumor cells can survive knockout of oncogenic KRAS and retain tumorigenic capacity likewise suggests that KRAS reprograms cells to a stably transformed state that no longer depends on continuous KRAS expression. However, oncogenic KRAS plays a profound immunosuppressive role in cancer development and maintenance.

Molecular mechanisms that control cancer immune escape still need to be investigated, but the described experiments give important clues on this point. While previous strategies to block KRAS oncogene therapeutically have focused on counteracting its growth-promoting role in cancer, we show that oncogenic KRAS plays a profound immunosuppressive role in cancer maintenance. Equally important is that we identify BRAF as a key regulator of PDAC maintenance and immune suppression. Based on the observed synergy between BRAF and MYC, we surmise that distinct MYC targets may enhance KRAS-induced tumor growth via reversible immune suppression. We also demonstrate that tumor tumor-intrinsic factors such as SMAD4 act as a barrier against tumor immune escape. As reported, growth of KRAS mutant pancreatic cancer cells with targeted inactivation of PIK3CA, a parallel branch of the KRAS signaling pathway, is also inhibited in syngeneic wild-type mice but not in nude mice due to the upregulation of MHC class I expression[58]. The results imply that the ability of mutant KRAS to modulate tumor immunity (via involvement of the RAF, PI3K, and MYC pathways) is an essential component of its oncogenicity, and that treatment of cancer will be improved by different modalities acting to simultaneously inhibit KRAS and activate immune pathways suppressed by cancer. Recent development of KRAS isoform-specific inhibitors highlights these expectations, as KRAS[G12C] blocker was found to drive antitumor immunity in a lung cancer model[7].

Our data show that KRAS ablation increases the influx of immune cells into the tumor, but with higher immune checkpoint expression that may be able to suppress the adaptive T-cell response. Accordingly, there are two issues that affect the outcome: the amount and composition of immune infiltrate, and the level of immune activation. Our study shows that the amount of immune infiltration increases upon KRAS ablation/inactivation, potentially creating a suitable milieu for a productive and robust antitumor response. The role of TGF-beta/SMAD4 in governing tumor cell antigenicity and immune signaling is of particular interest. Current models posit that RAS attenuates TGF-beta signaling, while TGF-beta stimulates the RAS pathway. In addition, TGF-beta has an adverse effect on tumor immunity and significantly inhibits host tumor immune surveillance[59,60]. Clearly, these models appear to contradict one another. The most obvious contradiction is that human PDACs contain mutations in SMAD4 and TGFBR2 genes in 35–50% of cases[20,26,61]. Losing one of these genes accelerates KRAS-induced carcinogenesis and enhances metastasis in mice[27,28]. These observations are not readily reconciled with the idea that tumors with altered TGF-beta/SMAD4 signaling have increased TGF-beta expression in their tumor-associated stroma[27]. How SMAD4 may affect T-cell recruitment beyond the known immunomodulatory effects of TGF-beta signaling remains unclear. Notably, recent evidence indicates that inhibition of TGF-beta can overcome resistance to checkpoint therapy[62,63]. Given this complexity, determining the role of individual genes in the TGF-beta pathway and KRAS tumor immunity will require further research.

## Methods

**Mammalian cells and reagents.** The KRAS[G12D] p53KO cell lines (KC), KRAS[G12D] p53[R172H] cell lines (KPC) (FC1199, FC1242, FC1245), and iKRAS p53[R172H/+] cell lines (A9312 and A9993) were described[2,19,21]. Cell lines were validated by Sanger sequencing to confirm KRAS mutation at the genomic level. KC cells were grown on gelatinized plates in CnT medium (CellnTec) supplemented with 1x antibiotic/antimycotic (Gibco), while KPC and iKRAS cell lines were grown in DMEM supplemented with 5% FBS. For proliferation assays, cells were seeded in 6-well plates ($4 \times 10^5$ cells per well) and counted with a Coulter counter for each time point[64]. Cell viability was measured using propidium iodide (PI) staining. Retroviral vectors were purchased from Addgene. For CRISPR/Cas9-mediated knockouts, we used sgKRAS RNA (5′-gtggttggagctgatggcgt-3′)[14], sgSMAD4 RNA (5′- ggtggcgttagactctgccg-3′)[61], and sgBRAF RNAs (5′-tcataattaacacacatcag-3′) and (5′-acaaatgattaagttgacac-3′)[50] cloned into LentiCRISPRv2. Recombinant viruses were produced by transient transfection of 293T or Phoenix cells according to standard protocols.

**Tumorigenicity in mice.** All animal studies have complied with all relevant ethical regulations for animal testing and research and were approved by the Institutional Animal Care and Use Committee at Stony Brook University (protocol 2011-0356). We used NU/J, C57BL/6J, FVB/NJ, CD4 KO, and CD8 KO mice (The Jackson Laboratory). Experimentally naive, adult male and female mice (8–10 weeks old) were housed individually, except for a 2-week period prior to surgery. SC and tail vein injections were performed with $10^4$ cells in 100 μl of Matrigel (diluted 1:10 with Opti-MEM). Orthotopic injections into the pancreas of mice were performed using standard procedures[65]. The animals were observed for tumor development by palpation. The endpoint was tumor diameter of 1 cm. Metastases was quantified by visual observation, histology, and bioluminescence imaging of the resected organs. Doxycycline-inducible KRAS[G12D] tumors were generated as described[2]. Anti-PD1 (BE0146) and anti-CTLA4 (BE0164) antibodies (BioXCell) were given at a dose of 80 μg each by IP injection on days 3, 6, and 9 after doxycycline withdrawal. For flow cytometry, tumor cells were dissociated with collagenase/hyaluronidase (Stem Cell Technologies). Live cells were scored using propidium iodide exclusion and stained with FITC, PE, or APC-conjugated antibodies to CD3, CD4, CD8, CD11B, CD19, CD45RA, and NK1.1 (eBioscience), and analyzed using FACSCalibur (BD) with CellQuest software. Frequency of tumor-initiating cells was determined using ELDA software (http://bioinf.wehi.edu.au/software/elda/).

**Immunostaining.** Mouse tissue was harvested and processed as described[19]. Paraffin-embedded 3-μm sections were processed and stained with hematoxylin and eosin. The slides were scored by two investigators. The following antibodies were used for IHC: rabbit polyclonal anti-KRT17/19, SOX9 (Cell Signaling), ACTA2, P-ERK, CD3, CD4 (Biolegend), CD8, CD11B, CD45, and CD68 (Abcam), supported with services of HistoWiz (Brooklyn, NY) and Stony Brook University Research Histology Core.

**Expression analysis.** Western blotting was performed using whole-cell extracts prepared by lysing cells in buffer containing 10 mM TrisHCl, pH 7.4, 150 mM NaCl, 1 mM EDTA, 10% glycerol, 1% Triton X100, 40 mM NaVO4, 0.1% SDS, and 1x protease inhibitors (Roche). Western blots were imaged and quantified using Image Studio software (LI-COR). Reverse-phase protein array analysis was performed by the MD Anderson RPPA Core Facility. RPPA data analysis was carried out using publicly available data sets and the existing literature[22]. The heatmaps were generated using Heatmapper software with Pearson correlation and centroid linkage. For RNA isolation, cells were harvested with TRIzol reagent (Invitrogen). RNA sequencing of tumor cells with basic bioinformatics and statistical analyses were performed by Novogene Corp. The TCGA data were downloaded as z-scores from the cBioPortal (www.cbioportal.org).

**Single-cell RNA sequencing.** Pancreatic tumors derived from orthotopic transplantation of KRAS intact and KRAS-deficient cancer cells into syngeneic C57BL/6J mice were mechanically and enzymatically dissociated into single-cell suspensions, followed by microfluidic partitioning into nanoliter droplets containing barcoded mRNA capture beads (Single Cell 3′ Reagent Kit v2; 10× Genomics). Single-cell barcoded cDNA libraries were prepared according to the manufacturer's protocol and sequenced on Illumina HiSeq 4000s (Novogene; http://en.novogene.com). Sequencing data were processed and analyzed by the 10× Genomics Cell Ranger pipeline (version 3.0.1) and Loupe Cell Browser v3.0.1.

**Pancreatic cancer gene list development.** Pancreatic adenocarcinoma clinical data and expression profiles (TCGA, provisional) were downloaded from cBioPortal (http://www.cbioportal.org), along with additional tumor and clinical annotations. Tumor-immune data were used as described[20,34,40]. Tumor samples were classified as RAS dependent and RAS independent on RDI scores[13]. Samples with RDI >4 were classified as RAS dependent and <0 as RAS independent. For the mouse score, samples with KS >2.8 were classified as KRAS high and <0 as KRAS low. Source data are provided as a Source data file. Scatter plots and heatmaps were drawn using R (https://www.r-project.org) and gplots package. Gene Set enrichment analysis was performed using the application available from the Broad Institute (http://software.broadinstitute.org/gsea/).

**Statistics and reproducibility**. Statistical analysis was performed using two-tailed Student's t-test, ANOVA analysis, Fisher's exact test, or Wilcoxon test, as appropriate for the dataset. An FDR-adjusted p-value (q-value) was calculated for multiple comparison correction. Data were analyzed using the R Project for statistical computing (https://www.r-project.org). Individual mice and tumor cell lines were considered biological replicates. Statistical details for each experiment are denoted in the corresponding figures and figure legends. The micrographs (H&E and IHC images) represent at least three independent experiments. For the quantification of IHC, the number of fields is indicated and p values between two groups were determined using the two-tailed t-test at the 0.05 confidence interval. All data are presented as mean ± SD. In box and whisker plots, the middle line is plotted at the median, the upper and lower hinges correspond to the first and third quartiles, and the ends of the whisker are set at 1.5× IQR above the third quartile and 1.5× IQR below the first quartile (IQR, interquartile range or difference between the 25th and 75th percentiles).

**Reporting summary**. Further information on research design is available in the Nature Research Reporting Summary linked to this article.

## Data availability

Pancreatic adenocarcinoma clinical data and expression profiles (TCGA, provisional) were downloaded from cBioPortal (http://www.cbioportal.org). The RNA-Seq data generated in this study have been deposited in the GEO/SRA database under accession code GSE132582. The single-cell sequencing data are deposited in the GEO/SRA database under accession code GSE146694. Source data are available as a Source data file. The remaining data and information are available within the Article, Supplementary Information or available from the authors upon request.

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

## Acknowledgements

The authors would like to thank Zainab Vasi for the generation of YAP1 KO PDAC cell lines and Jean Rooney (Stony Brook University) for assistance with animal surgery. For gifts of cell lines we thank David Tuveson (Cold Spring Harbor Laboratory), Richard Lin (Stony Brook University), and Marina Pasca di Magliano (University of Michigan). We acknowledge technical services by the Research Histology Core Laboratory (Stony Brook University). This work was supported by NIH grants R01AI105114, R01CA236389, and the Carol M. Baldwin Breast Cancer Research Award to N.C.R., NCI grant R01CA21720602 to R.S.P., and the Catacosinos Cancer Research Award to O.P.

## Author contributions

O.P. designed the research; I.I., S.D., M.R., and O.P. performed the research; M.R. and S.P. contributed new reagents/analytic tools; O.P., J.L., M.J.H., N.C.R., and S.P. analyzed the data; O.P. wrote the paper.

## Competing interests

The authors declare no competing interests.
