## [Peer Review File · Nature Communications]

REVIEWER COMMENTS

Reviewer #1 (Remarks to the Author): with expertise in PDAC models and signaling

The current manuscript describes a novel role of oncogenic Kras mediating immune suppression in pancreatic cancer cells. The authors use pancreatic cancer cell lines derived from the KPC mouse model, and inactivate Kras using CRISPR/Cas9 recombination. They then show that cell growth is only minimally impacted in vitro and in immune compromised mice. In contrast, they describe growth reduction in immune competent mice. The authors use single cell sequencing to describe changes in the microenvironment of tumors lacking Kras, and show changes in histology consistent with a more undifferentiated phenotype. Overall, the authors conclude that oncogenic Kras mediates immune evasion in pancreatic cancer.

Comments:

- 1) A limitation of the current study is the use of transplanted cells. While generating spontaneous models might go beyond the scope of the current manuscript, the limitation should be acknowledged. Further, the authors should ensure that an immune reaction against Cas9 is not causing the lack of growth in immune competent mice.
- 2) Functional experiments, such as depletion of key immune cell subsets in immune competent mice to determine whether tumor growth is restored, would support the notion that an immune response is involved.
- 3) The observation that cells lacking Kras retain elevated levels of p-ERK is contrary to previous observations (Ying et al., Cell 2012), and should be explained.
- 4) The authors should clearly refer to the specific cell lines used in their experiments. Most KPC (and iKras*) cell lines have been published with specific names, and this information is important as different lines have different immunogenicity.
- 5) The characterization of the immune response by single cell sequencing is insufficiently explained. The authors should show what combination of markers was used to attribute cellular identities, and what markers were used to define M1 and M2 macrophages, and the different fibroblast subtypes. Further, single cell sequencing is not an ideal technique to measure cellular composition, and tissue-based approaches (multiplex IHC or co-IF) should be used to validate the notion that immune and fibroblast subsets are altered in different transplanted tumors (with or without Kras). Further, apCAFs have also been identified as mesothelial cells, and to this reviewer's knowledge there is no evidence that they derive from monocyte progenitors. The origin should either be proven or the text altered.
- 6) In the description of Figure 4, it is important to state that the different tumors are "putatively" dependent on oncogenic Kras, rather than dependent, as no experimental validation is provided or can be provided.

Reviewer #2 (Remarks to the Author): with expertise in oncogenic KRAS and immune regulation

In this study, Ischenko and colleagues show that the dependence on KRAS mutations for tumour growth is reduced in advanced stages of pancreatic ductal adenocarcinoma (PDAC). While KRAS-deficient cells were able to form tumours in immuno-deficient mice, they failed to evade the host immune system in syngeneic wildtype mice and triggered an anti-tumour response. KRAS ablation increased the production of cytokines, stimulated recruitment of lymphoid cells and influenced cytokine production as well as macrophages and fibroblasts activation. Moreover, the authors show that BRAF ablation blocks KRAS-driven tumour growth and immune escape, while SMAD4 promotes immuno-surveillance. BRAF mutations are rare (3%) in patients with advanced pancreatic ductal adenocarcinoma.

Overall the authors address here an interesting question and their study provides new insights into the KRAS-mediated immune modulatory effects in pancreatic cancer. Major concerns that should

be addressed are listed below.

Major comments:

- Figure 2D: the legend indicate that several mice were included in the analysis (4 and 6) - why are there no error bars in the bar diagrams?
- The authors show in Fig, 2F, G that the loss of KRAS disrupts the immune composition of tumours, particularly leading to the infiltration of T and B cells. The data rely on quantification by immunohistochemistry (IHC) staining, which is rather a qualitative than a quantitative method. Defining distinct immune cell populations would be better done by flow cytometric analysis.
- The authors should also investigate the effect of KRAS ablation on immune checkpoint molecules/ligands (i.e. PD-L1, PD-1, CTLA4, TIM3 etc.), since various reports show an association between KRAS and PD-L1 in cancer. The same would apply for the patient samples.
- In Fig. 2D, at what time points did the mice from Fig. 2A, B develop metastasis? How was the quantification of tumour metastases performed in Fig. 2D? This is also missing in the methods part.
- The single-cell transcriptome profiling of KRAS intact and KRAS KO tumours in Fig. 3 is rather descriptive. The authors report differential expression of cytokines in KRAS KO tumours that could possibly be implicated in immune cell recruitment (particularly IFN), but do not provide further validation of these observations.
- In page 14, the authors state that more than 50% of syngeneic mice transplanted with KRAS/SMAD4 KO cells failed to develop tumours at endpoint 10 weeks post-injection. (i) Fig. S5A, B does not correspond to this conclusion (?) (ii) Did the authors investigate this further? Was there any analysis performed to study this immuno-suppressive impact of KRAS in vivo?
- The authors state that combination treatment which targets KRAS signalling and boosts the immune system will be an effective strategy to treat PDAC. However, the authors have not investigated this further in their manuscript. Ideally, they would use their models and perform functional studies where they can treat the mice with KRAS specific inhibitors or KRAS signalling targets (i.e. BRAFi in their case) in combination with immunotherapeutic agents.
- In page 15, referring to Fig. 5C, the authors wrote "...accompanied by a reduction in both absolute and relative numbers of TILs". However, Fig. 5C left panel is histology and IHC staining of representative tumours, while right panel is probably the percentage of mice developing tumours? Where is the data regarding TILs?
- Fig. 5F, G, again the immune infiltrates analysis is comprised of expression data, without further validation.
- The study lack further investigation of human PDAC patient samples. If possible, further analysis of human PDAC tumours (used in Fig. 4) would be ideal. This includes validating the expression of relevant genes (i.e. SMAD4), characterising infiltrated immune cells by flow cytometry, analysis of T cells/other immune cells within tumours, and validation of the expression of chemokine genes the authors find of particular interest "CCL5, CXCL9, CXCL12".
- In the discussion (page 21), the authors state: "Molecular mechanisms that control cancer immune escape have still to be investigated, but the described experiments give important clues on this point in order to hint towards its translational potential". Whilst the study does indeed give clues and hints, the study lack a lot of quantitative investigations, mechanistic studies as well as further validation for translational potential. The authors could have investigated these further, rather than wording the discussion in this way.

Minor comments:

- The (n) number of mice/samples is lacking throughout the figures or figure legends. Please include them for further assessment.
- In bar graphs, particularly ones showing latency (days), individual data points should be shown, rather than/ in addition to means.
- Box plots should be defined further (For example: indicate the interquartile range with the central bar indicating the median and the whiskers indicating the range).
- Error bars in some graphs are missing (e.g. Fig. 2B where error bars are described in figure legends but not physically shown, 5C, 7D) and are sometimes not described in figure legends (i.e.

Fig. 5B).

- In general, most figure legends are strikingly brief, sometimes not informative enough and therefore need substantial modifications.
- For indication of p-values, the authors use asterisks in some figures, but also use exact p-values in other figures; this should remain consistent throughout the manuscript. The definition of asterisks representing p-values is also missing in certain figure legends (e.g. Fig. 5B, 7E).
- Fig 1A is a confirmation of KRAS gene editing and can be moved to supplementary figures.
- Figure legends 1F, G, ideally the legends should be separated from each other.
- Fig. 4D, 5C, 7E, S1C data should be split into two panels in each figure.
- Fig. 5C (right panel - % of mice) was not described neither in the manuscript nor in the figure legends. Is it the percentage of mice that developed tumours?
- In page 15, Fig. S3D, E following up on Fig. 5, why are these figures in S3 rather than S5?
- Fig. 6C, the figure legend describing "mean latencies" does not correspond to the figure showing tumour initiating cell frequency.
- The "representative" western blot images and their perspective figure legends lack further information on how many experiments/samples the images represent.
- The methods are missing a statistical analysis part. Only one sentence in the "Tumorigenicity in mice" part of mentioned using student's t-test.
- "KWT" is wildtype, but is not defined neither in the manuscript text or figure legends.
- In page 7, the authors comment saying they did not observe changes in total or phosphorylated YAP1, and that YAP1 knockout in KRAS KO cells did not affect ERK phosphorylation. Where are these observations shown in the manuscript? Similarly in the following paragraph, the authors say they observed no effects of the proliferative properties of KRAS KO cells due to SMAD4 loss, which is also not shown in the figures.
- In certain figures (i.e. Fig. 5), the authors label the mice as "nude" and "wildtype", while in other figures (i.e. Fig. 6), they are labelled as "Foxn1nu" and "C57BL/6". This should remain consistent throughout the manuscript.
- A number of orthographical errors can be found throughout the manuscript.

Reviewer #3 (Remarks to the Author): with expertise in PDAC, scRNAseq and transcriptomic

Ischenko et al. have used CRISPR to inactivate KRAS in a PDAC mouse models leads to an increased immune response against the tumor compared to mutant KRAS driven PDAC. The authors use the KRASG12D p53KO mouse cell line (KC) and delete KRASG12D by CRISPR/CAS9. The authors show early KC neoplastic lines are unable to survive without KRASG12D, while established tumors had a higher rate of colony formation after KRASG12D CRISPR. The KRAS KO lines are able to form tumors and metastasize. Interestingly, KO lines had a poorly differentiated morphology and increased mesenchymal gene expression (ACTA2). Along with EMT signatures and TGF-beta signaling. SMAD4 deletion partially abrogates the gain in EMT phenotype in KRAS KO cells. Interestingly, KRAS KO cells in a syngeneic mouse appear to have lower tumor forming ability suggestive of immune surveillance differences between KRAS KO and mutant cells. This was confirmed with a KRASG12D dox inducible system. The authors move on to evaluate single cell RNA-seq analysis of tumors defining epithelial (E) and basal/mesenchymal (M) cancer cells and iCAFs and myCAFs. They note M1, M2, and TAMs with mixed M1/M2 phenotype in these tumors. KRAS KO tumors had significant shifts to EMT phenotypes with ERK activation in tumor cells. Interestingly, they found increased MHC Class I and Class II expression in tumor cells, MHC class II apcCAFs, TAMs were M1 polarized, TILs dominated by CD8 T and NKT cells and associated IFN gamma production. Using a KRAS mutant cell line panel and scRNA-seq signatures they make a KRAS-dependency score and find KRAS dependent E and M tumors (KE and KM) as well as KRAS independent tumors (KRAS-low) in TCGA datasets. KM and KRAS-low tumors had higher T-cell infiltration based on digital image analysis of the TCGA slides. They then move into BRAF and MYC over-expression as a rescue with accelerated tumor formation in nude mice, but neither of these genes were able to rescue tumorigenesis in WT mice indicating that they are not sufficient to

rescue from immune surveillance. However, BRAF and MYC was able to reproduce morphology of KRAS intact tumors. They then show BRAF appears to be important in KRAS driven tumors and KO of BRAF inhibits tumor formation in WT mice. Finally the authors show that SMAD4 loss in BRAF/MYC tumors to nearly completely rescue for KRAS KO. Overall an impressive amount of collective work shown in this single manuscript. However, the relevance of this to human tumors is not clear given that EMT PDAC (Basal/QM/Squamous) is known to be more aggressive than epithelial and the immune infiltrate is therefore not effective. This would suggest that KRAS inhibition/deletion driving an EMT PDAC would be detrimental with increased metastatic potential despite enhanced immune response.

Major comments/questions

1. Given the multiple manipulations described, it might be useful to have an overall model in the final figure.
2. How do the authors reconcile EMT PDAC having a higher immune infiltrate (Fig 4), but in general we know these patients often have worsened survival?
3. Based on their analysis the higher immune cells in KM tumors have high PDL1 (CD274) and CTLA4 indicating that T-cells are suppressed by these two checkpoints. However, we know combined checkpoint in PDAC has not been effective. Do the authors have any potential ideas of why higher immune infiltrates do not correlate to better response to immunotherapy? Does this CD274 and CTLA4 elevation also occur in the mouse models of KRAS KO?
4. Have the authors looked at metastatic potential of KRAS KO vs MUT cells via tail vein or orthotopic tumors? How about migration/invasion assays? Figure 1D would indicate that Tumor initiating capability is higher in KO. This finding would be very interesting and provocative since it would be counter to the dogma that targeting KRAS is universally a good idea...

Minor comments

1. Page 11 "MK167" I think should be MKI67
2. Page 12, I think you want Natural Killer T to be "NKT"
3. Page 22 "DMAD4" should be SMAD4

Reviewer #4 (Remarks to the Author): with expertise in PDAC and immune regulation

This is an interesting article that elaborates on the mechanism of immune evasion in the context of KRAS in pancreatic cancer. Although a very relevant topic and interesting approach, the manuscript has numerous significant drawbacks in data presentation/interpretation and experimental set ups as detailed below.

- 1) What is the control cell? Is it the parental batch cellular population without CRISPR lentiviral transduction, or transduced with empty pLentiCRISPR vector in figure 1A.
- 2) For the generation of metastatic, tumor and metastatic cells, the authors refer back to their previous publication (ref 20). However, a clearer description in this manuscript is essential.
- 3) Figure 1C: please indicate how many mice are transplanted by how many clones? Does each line indicate a single cell with multiple mice transplantation? Does this mean that you have only one control cell? Is this control cell the parental untransduced KC cell or transduced but not Kras crisped cell line? If it is not the parental cell line, it would be nice, if the authors add at least one more control cell clones.
- 4) Figure 1D: in the main text transplanted tumor cells are told to be both KC/KPC. However, in the figure legend it is said to be KPC. Please clarify.
- 5) In supplementary figure 1C, please include the LN metastasis picture from also the KRAS intact cells from tail vein. A higher magnification of the metastatic sites would be nice to understand whether they also had morphological/histological differences as it was observed in primary tumors (subcutaneous).
- 6) Overall for figure 1 and supplementary figure 1:
The authors refer to Kras KO cells (n=10) to be both KC and KPC cells. In the individual figures,

either of the genotypes are used. For each experiment, rather than saying KRAS KO, please indicate whether KC or KPC is used.

It is not clear what exactly n number refers to. Did authors transplant different control and KRAS KO cells in to different mice, or did they use same cell line for each transplantation? I can understand that even if the cell line is same, different mice means biological replicates. However, I am sceptical with the use of single control cell especially in subcutaneous transplantation experiment in figure 1C. Is it also the case in Supplementary figure 1C?

7) Please include both macroscopic and lower magnification microscopic pictures of the tumors. The number of histological type for both intact and knock out mice should also be included. How many mice among how many had sarcomatous tumor? Please also include the KRT19 and ACTA2 IHC pictures or the KRAS intact cohort.

8) In supplementary figure 1D, I still observe a difference in p-ERK levels in both pretumor and tumor cell lines. However, this has not been mirrored in RNAseq and RPPA analysis in figure S1E and S1F. What is the insight of the authors about this? Would it still remain same, if they used multiple control cells in the blots? Especially, p-AKT levels look very different in KRAS intact and KO cells which is not mirrored in RNAseq and RPPA analysis. How about the p-AKT levels in tumor cells (S1D).

9) Authors mention about p-YAP1 mediate p-ERK rescue. However, the experimental data is not provided.

10) Is figure 1I subcutaneous transplantation latency?

11) In figure 2B, please indicate how many different cell clones are transplanted to how many different mice. Again, is there only one single intact cell line which is transplanted into 10 different mice?

12) The metastasis data is quite interesting in figure 2D. A graph depicting % metastasis in mice which formed tumor would be better. Although the number of mice with tumor is low, it seems they develop metastasis with very high efficiency. How about tail vein injection of the KRAS intact/KO KPC cells into syngeneic mice? Would immune system still suppress primary foci formation in a Kras dependent manner in lung and Lymph nodes metastatic area

13) Figure S2B-C: Can iKras cells also form tumor in a different schedule. If so, would Doxycyclin withdrawal also impact metastatic foci? My overall question is how important is maintained Kras signaling also for metastatic foci? Can Kras therapies have an impact also on metastatic sites? The authors have great tools to answer these questions? Could they provide also an immune cell staining panel for the metastatic tumors of KRAS intact and WT mice?

14) The results with the MHC expression is quite intriguing. The researchers state that MHC dependent antigen presentation impact in immune response is arguable. However, recent publications indicate how important MHC presentation can be for both primary and metastatic tumors (e.g. Yamamoto et. Al. 2020, Autophagy promotes immune evasion of pancreatic cancer by degrading MHC-I, Pommier et al. Unresolved endoplasmic reticulum stress engenders immune-resistant, latent pancreatic cancer metastases. Science (2018)). From their RNAseq or RPPA results, did authors find any enriched pathways regarding the autophagy, ER stress mediated MHC expression regulation?

15) Figure 4/S4: Is KM signature correlated with both of the Ras dependency signatures? Authors only mention about the link in mutations observed in Kras and other inactivating tumor suppressor genes. Kras dependency difference in KM and KE tumors of TCGA is not nicely demonstrated in their analysis. Are Kras independent tumors KM tumors in figure 4a? Otherwise, just because SMAD4/2 expression is linked, can they state KM samples as Kras independent?

16) Manuscripts subtyping pancreatic cancer based on transcription signatures indicate that tumors with EMT signatures (squamous, basal, quasimesenchymal etc...) have immune evasion phenotype or less immune cell infiltration. However, results here indicate an opposite output. Do they think a Kras dependency stratification of basal/squamous/quasimesenchymal type PDAC samples would distinctively divide immune cell phenotype? Does Kras dependency/activity results in a difference in Kaplan meier plot in TCGA samples??

17) Would antibody mediated CD8+ T cell depletion fully allow KRAS KO tumors to grow in syngeneic transplanted tumors?

- 18) What happens to other RAS isoforms when KRAS is knocked out?
- 19) Total Ras activity assay is required in KRAS intact vs KO mice.

Point by point response

Reviewer #1 (Remarks to the Author): with expertise in PDAC models and signaling

The current manuscript describes a novel role of oncogenic Kras mediating immune suppression in pancreatic cancer cells. The authors use pancreatic cancer cell lines derived from the KPC mouse model, and inactivate Kras using CRISPR/Cas9 recombination. They then show that cell growth is only minimally impacted in vitro and in immune compromised mice. In contrast, they describe growth reduction in immune competent mice. The authors use single cell sequencing to describe changes in the microenvironment of tumors lacking Kras, and show changes in histology consistent with a more undifferentiated phenotype. Overall, the authors conclude that oncogenic Kras mediates immune evasion in pancreatic cancer.

Comments:

1) A limitation of the current study is the use of transplanted cells. While generating spontaneous models might go beyond the scope of the current manuscript, the limitation should be acknowledged. Further, the authors should ensure that an immune reaction against Cas9 is not causing the lack of growth in immune competent mice.

- This limitation has been acknowledged, as requested (Discussion, p. 21).
- There isn't an ideal approach. Cre-mediated excision of mutant KRAS in vivo is limited, as it would have to occur with 100% efficiency. The inducible KRAS system is a reasonable approximation, but >50% of tumors fail to regress for various reasons (DePinho et al., 2014).
- We used Cas9-expressing controls throughout the study, as stated in the text (p. 8).

2) Functional experiments, such as depletion of key immune cell subsets in immune competent mice to determine whether tumor growth is restored, would support the notion that an immune response is involved.

- This was done as requested. C57Bl/6 mice lacking CD4 or CD8 T cells fail to reject KRAS KO tumors (Fig. S2c).

3) The observation that cells lacking Kras retain elevated levels of p-ERK is contrary to previous observations (Ying et al., Cell 2012), and should be explained.

- The observation is not new. Studies demonstrate activation of the MAPK/ERK pathway in the absence of RAS genes (M. Barbacid, PNAS, 2014; T. Jacks, Nature Comm., 2017). The non-canonical modes of MAPK/ERK activation are well acknowledged. The consensus is that under regular growth conditions, cells lacking KRAS closely resemble KRAS intact controls.
- In fact, endogenous RAS mutants only moderately elevate ERK activity (F. McCormick's studies). The conversion of a wild type KRAS gene to an activating KRAS mutant can actually reduce pERK levels, despite inducing tumor formation (Tuveson et al., Cancer Cell, 2004).
- The doxy-inducible KRASG12D/p53KO transgene used in the referenced study (Ying et al., 2012) is driven by an exogenous (CMV) promoter. It is marked by high levels of KRASG12D expression, but a non-metastatic phenotype. Our doxy-inducible KRASG12D/p53R172H transgene (Collins et al., 2012) is devoid of these shortcomings.

4) The authors should clearly refer to the specific cell lines used in their experiments. Most KPC (and iKras*) cell lines have been published with specific names, and this information is important as different lines have different immunogenicity.

- We added a detailed description of cell lines (Fig. S1b).

5) The characterization of the immune response by single cell sequencing is insufficiently explained. The authors should show what combination of markers was used to attribute cellular identities, and what markers were used to define M1 and M2 macrophages, and the different fibroblast subtypes. Further, single cell sequencing is not an ideal technique to measure cellular composition, and tissue-based approaches (multiplex IHC or co-IF) should be used to validate the notion that immune and fibroblast subsets are altered in different transplanted tumors (with or without Kras). Further, apCAFs have also been identified as mesothelial cells, and to this reviewer's knowledge there is no evidence that they derive from monocyte progenitors. The origin should either be proven or the text altered.

- Tumor cells were scored and clustered as described (Moffitt et al., Nature genetics. 2015; Roe et al., Cell. 2017). Classical tumors exhibit elevated expression of epithelial genes (e.g., CDH1, EPCAM, keratin genes) and transcription factors involved in pancreatic differentiation (FOXA1, GATA6, HNF4A, ONECUT2). Basal-like tumors exhibit activation of EMT (ACTA2, FN1, collagen genes) as well as FOX, SOX and ZEB transcription factor families (Fig. 3c).

- Stromal cells were clustered as described (Öhlund et al. J Exp Med. 2017; Elyada et al., Cancer discovery, 2019). Myofibroblasts exhibit high ACTA2 and THY1 expression; inflammatory CAFs exhibit high levels of cytokines (IL6, CXCL2, etc.); and antigen-presenting CAFs show high CD74 and MHC class II expression (Fig. 3d).
- Markers used to define M1 and M2 macrophages are now shown in Fig. S3a. They were clustered as described (Azizi et al., Cell. 2018).
- Mentioning of monocytes has been removed.
- We acknowledge the value of full and unbiased characterization of transplanted tumors in their proper histological context. However, one has to bear in mind that KRAS KO tumors are difficult to produce in wild-type mice, as ~90% of them are rejected. Tumors that are rejected are untestable. An in-depth analysis of a minority of escapees would take away from the key focus of the study (the deleterious effects of KRAS loss). In these circumstances, our preferred approach was to validate the genotype-phenotype relations by targeted gene expression. For our purposes, this approach is more instructive.

6) In the description of Figure 4, it is important to state that the different tumors are “putatively” dependent on oncogenic Kras, rather than dependent, as no experimental validation is provided or can be provided.

- This was done as requested.

Reviewer #2 (Remarks to the Author): with expertise in oncogenic KRAS and immune regulation

In this study, Ischenko and colleagues show that the dependence on KRAS mutations for tumour growth is reduced in advanced stages of pancreatic ductal adenocarcinoma (PDAC). While KRAS-deficient cells were able to form tumours in immuno-deficient mice, they failed to evade the host immune system in syngeneic wildtype mice and triggered an anti-tumour response. KRAS ablation increased the production of cytokines, stimulated recruitment of lymphoid cells and influenced cytokine production as well as macrophages and fibroblasts activation. Moreover, the authors show that BRAF ablation blocks KRAS-driven tumour growth and immune escape, while SMAD4 promotes immuno-surveillance. BRAF mutations are rare (3%) in patients with advanced pancreatic ductal adenocarcinoma.

Overall the authors address here an interesting question and their study provides new insights into the KRAS-mediated immune modulatory effects in pancreatic cancer. Major concerns that should be addressed are listed below.

Major comments:

- Figure 2D: the legend indicate that several mice were included in the analysis (4 and 6) - why are there no error bars in the bar diagrams?

- The graph shows percentage of mice with metastases. Error bars are not appropriate.

- The authors show in Fig, 2F, G that the loss of KRAS disrupts the immune composition of tumours, particularly leading to the infiltration of T and B cells. The data rely on quantification by immunohistochemistry (IHC) staining, which is rather a qualitative than a quantitative method. Defining distinct immune cell populations would be better done by flow cytometric analysis.

- Flow cytometry has been added as requested (Fig S2e). However, one has to bear in mind that KRAS KO tumors are difficult to produce in wild-type mice, as most of them are rejected. Quantitation of escapees (~10%) leads to an altogether different question. This is why we chose IHC as the major method.

- The authors should also investigate the effect of KRAS ablation on immune checkpoint molecules/ligands (i.e. PD-L1, PD-1, CTLA4, TIM3 etc.), since various reports show an association between KRAS and PD-L1 in cancer. The same would apply for the patient samples.

- We did investigate this matter. Data are shown for both mouse and human tumors (Fig. S3c and Fig. 4c, g).

- In Fig. 2D, at what time points did the mice from Fig. 2A, B develop metastasis? How was the quantification of tumour metastases performed in Fig. 2D? This is also missing in the methods part.

- This information has been added to the methods part (p. 25).

- The single-cell transcriptome profiling of KRAS intact and KRAS KO tumours in Fig. 3 is rather descriptive. The authors report differential expression of cytokines in KRAS KO tumours that could possibly be implicated in immune cell recruitment (particularly IFN), but do not provide further validation of these observations.

- We emphasize increased expression of CCL5, CXCL9, CXCL10, and MHC class II genes following loss of KRAS. Abundant evidence indicates that the combined expression of these genes is associated with improved patient survival. The results are based on the analysis of over 10,000 tumors comprising 33 cancer types (The Immune Landscape of Cancer. *Immunity*, 2018). The biological effects of CCL5, CXCL9, and CXCL10 on immune cell recruitment have been extensively studied (Dangaj et al. *Cancer Cell*. 2019). The point here is that there is a wealth of knowledge and evidence to draw upon. We referenced these studies.
 - Another important point is that IFNG has no chemotactic activity. This is precisely why uncovering the genetic drivers of tumor immunity takes precedence over genetic markers of tumor immunity, such as IFNG.
- In page 14, the authors state that more than 50% of syngeneic mice transplanted with KRAS/SMAD4 KO cells failed to develop tumours at endpoint 10 weeks post-injection. (i) Fig. S5A, B does not correspond to this conclusion (?) (ii) Did the authors investigate this further? Was there any analysis performed to study this immuno-suppressive impact of KRAS in vivo?
- Fig. S5A is in accord with the rest of the study. KRAS/SMAD4 KO cells are referred to as “weakly tumorigenic” in wild-type mice.
 - We investigated this matter in detail in Fig. 7, including the role of SMAD4 in tumor growth, histology and immune infiltration.
- The authors state that combination treatment which targets KRAS signalling and boosts the immune system will be an effective strategy to treat PDAC. However, the authors have not investigated this further in their manuscript. Ideally, they would use their models and perform functional studies where they can treat the mice with KRAS specific inhibitors or KRAS signalling targets (i.e. BRAFi in their case) in combination with immunotherapeutic agents.
- Checkpoint inhibition has been added as requested (new Fig S2k).
 - The request regarding inhibition of KRAS signaling cannot be reasonably accomplished. Currently, there are no pancreas-specific KRAS inhibitors (G12D, G12V or G12R) or pan-KRAS inhibitors. There are no usable inhibitors of RAF (M. Barbacid et al., *Cancer Cell*. 2019). RAF inhibitors categorized as a “BRAF” inhibitors suppress RAF activity almost exclusively in BRAF-mutant cells. Other therapeutic agents, e.g. MEK inhibitors, suppress T cell function (Cannon et al., *Nature*. 2019).

- In page 15, referring to Fig. 5C, the authors wrote “...accompanied by a reduction in both absolute and relative numbers of TILs”. However, Fig. 5C left panel is histology and IHC staining of representative tumours, while right panel is probably the percentage of mice developing tumours? Where is the data regarding TILs?

- BRAF/MYC tumors are devoid of TILs (Fig. S5b).

- Fig. 5F, G, again the immune infiltrates analysis is comprised of expression data, without further validation.

- The aim of these experiments involving BRAF and MYC was to test the effect of BRAF and MYC on T cell infiltration of the tumor. We used two orthogonal approaches, single cell sequencing and IHC (Fig. S5b, e).

- The study lack further investigation of human PDAC patient samples. If possible, further analysis of human PDAC tumours (used in Fig. 4) would be ideal. This includes validating the expression of relevant genes (i.e. SMAD4), characterising infiltrated immune cells by flow cytometry, analysis of T cells/other immune cells within tumours, and validation of the expression of chemokine genes the authors find of particular interest “CCL5, CXCL9, CXCL12”.

- Our data in Fig. 4 are based on computational genomics and digital pathology images from the TCGA. The power of these approaches extends beyond what can be achieved with flow cytometry. Reexamining TCGA samples by FACS is not feasible. TCGA has no rights to redistribute materials outside of the program.
- To address the reviewer’s concern, we reexamined the TCGA database. In brief, human PDACs (n=136) contain a measurable number of lymphoid cells and macrophages, but little or no neutrophils and dendritic cells (new Fig. S4f).
- According to TCGA immune classification (Immunity, 2018), KRAS-high tumors fall into four categories: C1 (termed wound healing), C2 (IFNG dominant), C3 (inflammatory) and C6 (TGF-beta dominant). In contrast, KRAS-low tumors are significantly skewed towards the C3 subtype (p=0.003) (new Fig. 4f). This subtype is defined by a high Th1/Th2 ratio (in accordance with our data in Fig. 3), low to moderate tumor cell proliferation, and the most favorable prognosis.

- In the discussion (page 21), the authors state: “Molecular mechanisms that control cancer immune escape have still to be investigated, but the described experiments give important clues on this point in order to hint towards its translational potential”. Whilst the study does indeed

give clues and hints, the study lack a lot of quantitative investigations, mechanistic studies as well as further validation for translational potential. The authors could have investigated these further, rather than wording the discussion in this way.

- Our study opens up a new avenue of experimental investigation. We show experimentally that the ability of KRAS to evade the immune system is an essential component of its oncogenicity. It is an achievement on its own. Mechanistically, we demonstrate that ablation of BRAF effectively blocks KRAS-dependent cancer growth. It is a major result of high translational value.

Minor comments:

- The (n) number of mice/samples is lacking throughout the figures or figure legends. Please include them for further assessment.
 - This was done.
- In bar graphs, particularly ones showing latency (days), individual data points should be shown, rather than/ in addition to means.
 - This was done.
- Box plots should be defined further (For example: indicate the interquartile range with the central bar indicating the median and the whiskers indicating the range).
 - This was done.
- Error bars in some graphs are missing (e.g. Fig. 2B where error bars are described in figure legends but not physically shown, 5C, 7D) and are sometimes not described in figure legends (i.e. Fig. 5B).
 - The percentages are shown. Error bars are not appropriate.
- In general, most figure legends are strikingly brief, sometimes not informative enough and therefore need substantial modifications.
 - We have addressed this point.

- For indication of p-values, the authors use asterisks in some figures, but also use exact p-values in other figures; this should remain consistent throughout the manuscript. The definition of asterisks representing p-values is also missing in certain figure legends (e.g. Fig. 5B, 7E).
 - This has been corrected.
- Fig 1A is a confirmation of KRAS gene editing and can be moved to supplementary figures.
 - We chose to keep this figure.
- Figure legends 1F, G, ideally the legends should be separated from each other.
 - This was done.
- Fig. 4D, 5C, 7E, S1C data should be split into two panels in each figure.
 - This was done.
- Fig. 5C (right panel - % of mice) was not described neither in the manuscript nor in the figure legends. Is it the percentage of mice that developed tumours?
 - This was corrected.
- In page 15, Fig. S3D, E following up on Fig. 5, why are these figures in S3 rather than S5?
 - This was corrected.
- Fig. 6C, the figure legend describing “mean latencies” does not correspond to the figure showing tumour initiating cell frequency.
 - This was corrected.
- The “representative” western blot images and there perspective figure legends lack further information on how many experiments/samples the images represent.
 - Each lane in the western blot represents a lysate from a stable cell line that we generated. They went into 6-8 mice. The Source data file provides the details.
- The methods are missing a statistical analysis part. Only one sentence in the “Tumorigenicity in mice” part of mentioned using student’s t-test.
 - Statistical tools are included in the Data Availability section.

- “KWT” is wildtype, but is not defined neither in the manuscript text or figure legends.
 - This was corrected.
- In page 7, the authors comment saying they did not observe changes in total or phosphorylated YAP1, and that YAP1 knockout in KRAS KO cells did not affect ERK phosphorylation. Where are these observations shown in the manuscript? Similarly in the following paragraph, the authors say they observed no effects of the proliferative properties of KRAS KO cells due to SMAD4 loss, which is also not shown in the figures.
 - Yap1 data are now shown in Fig S1i.
- In certain figures (i.e. Fig. 5), the authors label the mice as “nude” and “wildtype”, while in other figures (i.e. Fig. 6), they are labelled as “Foxn1nu” and “C57BL/6”. This should remain consistent throughout the manuscript.
 - This was corrected.
- A number of orthographical errors can be found throughout the manuscript.
 - We have addressed this point.

Reviewer #3 (Remarks to the Author): with expertise in PDAC, scRNAseq and transcriptomic

Ischenko et al. have used CRISPR to inactivate KRAS in a PDAC mouse models leads to an increased immune response against the tumor compared to mutant KRAS driven PDAC. The authors use the KRASG12D p53KO mouse cell line (KC) and delete KRASG12D by CRISPR/CAS9. The authors show early KC neoplastic lines are unable to survive without KRASG12D, while established tumors had a higher rate of colony formation after KRASG12D CRISPR. The KRAS KO lines are able to form tumors and metastasize. Interestingly, KO lines had a poorly differentiated morphology and increased mesenchymal gene expression (ACTA2). Along with EMT signatures and TGF-beta signaling. SMAD4 deletion partially abrogates the gain in EMT phenotype in KRAS KO cells. Interestingly, KRAS KO cells in a syngeneic mouse appear to have lower tumor forming ability suggestive of immune surveillance differences between KRAS KO and mutant cells. This was confirmed with a KRASG12D dox inducible system. The authors move on to evaluate single cell RNA-seq analysis of tumors defining epithelial (E) and basal/mesenchymal (M) cancer cells and iCAFs and myCAFs. They note M1,

M2, and TAMs with mixed M1/M2 phenotype in these tumors. KRAS KO tumors had significant shifts to EMT phenotypes with ERK activation in tumor cells. Interestingly, they found increased MHC Class I and Class II expression in tumor cells, MHC class II apcCAFs, TAMs were M1 polarized, TILs dominated by CD8 T and NKT cells and associated IFN gamma production. Using a KRAS mutant cell line panel and scRNA-seq signatures they make a KRAS-dependency score and find KRAS dependent E and M tumors (KE and KM) as well as KRAS independent tumors (KRAS-low) in TCGA datasets. KM and KRAS-low tumors had higher T-cell infiltration based on digital image analysis of the TCGA slides. They then move into BRAF and MYC over-expression as a rescue with accelerated tumor formation in nude mice, but neither of these genes were able to rescue tumorigenesis in WT mice indicating that they are not sufficient to rescue from immune surveillance. However, BRAF and MYC was able to reproduce morphology of KRAS intact tumors. They then show BRAF appears to be important in KRAS driven tumors and KO of BRAF inhibits tumor formation in WT mice. Finally the authors show that SMAD4 loss in BRAF/MYC tumors to nearly completely rescue for KRAS KO. Overall an impressive amount of collective work shown in this single manuscript. However, the relevance of this to human tumors is not clear given that EMT PDAC (Basal/QM/Squamous) is known to be more aggressive than epithelial and the immune infiltrate is therefore not effective. This would suggest that KRAS inhibition/deletion driving an EMT PDAC would be detrimental with increased metastatic potential despite enhanced immune response.

Major comments/questions

1. Given the multiple manipulations described, it might be useful to have an overall model in the final figure.

- This was done in Fig. S6.

2. How do the authors reconcile EMT PDAC having a higher immune infiltrate (Fig 4), but in general we know these patients often have worsened survival?

- Pancreatic cancer expression profiles typically reflect a classical or basal-like phenotype (i.e. EMT PDAC). A recent de novo reclassification of these tumor types has been illuminating (Chan-Seng-Yue et al. Nature genetics, 2020; Hayashi et al. Nature Cancer, 2020). Previous 'basal-like' tumors are composed predominantly of mixed tumors of uncertain classification; basal-like and classical programs can coexist within the same

tumor; the basal-like type expression signature reflects squamous differentiation in PDACs.

- Squamous differentiation does is not related to EMT. However, adenosquamous carcinoma (basal-like molecular subtype) has intrinsic chemotherapy resistance. EMT is a manifestation of epithelial plasticity in basal tumors induced by cancer drugs (Bhang et al., Nature Med. 2015).
- To sum up, much of the knowledge about PDAC subtypes (basal/QM), survival and the immune resistance needs to be reassessed.
- Our data do not support a causative link between tumor morphology and immune infiltration, although both are related to KRAS loss. The key here is loss of KRAS expression rather than EMT. Patients without the KRAS mutation survive significantly longer than those with mutant KRAS.

3. Based on their analysis the higher immune cells in KM tumors have high PDL1 (CD274) and CTLA4 indicating that T-cells are suppressed by these two checkpoints. However, we know combined checkpoint in PDAC has not been effective. Do the authors have any potential ideas of why higher immune infiltrates do not correlate to better response to immunotherapy? Does this CD274 and CTLA4 elevation also occur in the mouse models of KRAS KO?

- TGF-beta has been identified as the principal barrier. Inhibition of TGF-beta by various means overcomes resistance to checkpoint therapy (e.g., Principe et al., Mol Cancer Ther. 2019; Martin et al. Sci Transl Med. 2020).
- CD274 and CTLA4 elevation occurs in KRAS KO tumors by virtue of increased immune infiltration (Fig. S3c).

4. Have the authors looked at metastatic potential of KRAS KO vs MUT cells via tail vein or orthotopic tumors? How about migration/invasion assays? Figure 1D would indicate that Tumor initiating capability is higher in KO. This finding would be very interesting and provocative since it would be counter to the dogma that targeting KRAS is universally a good idea...

- Both tail vein and orthotopic injections were performed (Fig. S1h and Fig. S2a). Loss of KRAS reduces the metastatic capacity of tumor cells (Fig. S2a).
- Figure 1d shows the opposite. Loss of KRAS reduces tumor initiating capability of cells.
- Targeting KRAS is a good idea. However, KRAS inhibition needs to be combined with immune activation.

Minor comments

1. Page 11 “MK167” I think should be MK167

- This was corrected.

2. Page 12, I think you want Natural Killer T to be “NKT”

- This was corrected.

3. Page 22 “DMAD4” should be SMAD4

- This was corrected.

Reviewer #4 (Remarks to the Author): with expertise in PDAC and immune regulation

This is an interesting article that elaborates on the mechanism of immune evasion in the context of KRAS in pancreatic cancer. Although a very relevant topic and interesting approach, the manuscript has numerous significant drawbacks in data presentation/interpretation and experimental set ups as detailed below.

1) What is the control cell? Is it the parental batch cellular population without CRISPR lentiviral transduction, or transduced with empty pLentiCRISPR vector in figure 1A.

- C stands for control parental cells. CRISPR-transduced control cells were used in orthotopic injections.

2) For the generation of metastatic, tumor and metastatic cells, the authors refer back to their previous publication (ref 20). However, a clearer description in this manuscript is essential.

- We added a detailed description of cell lines (new Fig. S1b).

3) Figure 1C: please indicate how many mice are transplanted by how many clones? Does each line indicate a single cell with multiple mice transplantation? Does this mean that you have only one control cell? Is this control cell the parental untransduced KC cell or transduced but not Kras crisped cell line? If it is not the parental cell line, it would be nice, if the authors add at least one more control cell clones.

- The graph is representative of >10 independent injections. New Fig. S1e provides the details.

- We used multiple control cell lines. New Fig. S1b provides the details.
- We used CRISPR/Cas9-transduced control cells, as indicated (Results, page 8).

4) Figure 1D: in the main text transplanted tumor cells are told to be both KC/KPC. However, in the figure legend it is said to be KPC. Please clarify.

- This particular experiment was performed using KPC cells, as indicated.

5) In supplementary figure 1C, please include the LN metastasis picture from also the KRAS intact cells from tail vein. A higher magnification of the metastatic sites would be nice to understand whether they also had morphological/histological differences as it was observed in primary tumors (subcutaneous).

- This is done in new Fig. S2b. Metastatic tumors have the same type of cells as the primary tumor, consistent with prior studies (Makohon-Moore et al., Nature genetics, 2017; Reiter et al., Science, 2018; Connor et al., Cancer Cell, 2019; Priestley et al., Nature, 2019).

6) Overall for figure 1 and supplementary figure 1:

The authors refer to Kras KO cells (n=10) to be both KC and KPC cells. In the individual figures, either of the genotypes are used. For each experiment, rather than saying KRAS KO, please indicate whether KC or KPC is used.

- This was corrected.

6) It is not clear what exactly n number refers to. Did authors transplant different control and KRAS KO cells in to different mice, or did they use same cell line for each transplantation? I can understand that even if the cell line is same, different mice means biological replicates. However, I am sceptical with the use of single control cell especially in subcutaneous transplantation experiment in figure 1C. Is it also the case in Supplementary figure 1C?

- KRAS intact cells were transplanted into >30 mice; KRAS KO cells were transplanted into >50 mice. Fig. S1e provides the details, including multiple controls.

7) Please include both macroscopic and lower magnification microscopic pictures of the tumors. The number of histological type for both intact and knock out mice should also be included. How

many mice among how many had sarcomatous tumor? Please also include the KRT19 and ACTA2 IHC pictures or the KRAS intact cohort.

- New images are shown in Fig. S2b, as requested.
- KRAS intact cell lines produced moderately to well-differentiated adenocarcinomas, while KRAS KO cell lines invariably generated poorly differentiated tumors. The observation is not new. Similar morphological changes were observed before (T. Jacks. Nature Comm., 2017; Mou et al., PNAS, 2017).
- KRT19 and ACTA2 staining has been presented (Fig. S1j).

8) In supplementary figure 1D, I still observe a difference in p-ERK levels in both pretumor and tumor cell lines. However, this has not been mirrored in RNAseq and RPPA analysis in figure S1E and S1F. What is the insight of the authors about this? Would it still remain same, if they used multiple control cells in the blots? Especially, p-AKT levels look very different in KRAS intact and KO cells which is not mirrored in RNAseq and RPPA analysis. How about the p-AKT levels in tumor cells (S1D).

- The ability of cells to partially or fully compensate for the loss of KRAS in terms of survival, growth and ERK activation is well documented (M. Barbacid, PNAS. 2014; T. Jacks, Nature Comm. 2017). The point of our Fig. 1 is not MAPK, but the demonstration that the tumorigenic defects of KRAS KO cells depend on SMAD4.
- The pathway score is defined as a summation of the weights of individual genes in the pathway. We analyzed RPPA data using a scoring method of Akbani et al., which has been adopted by the TCGA (Nature Commun, 2014). The list includes a total of 10 proteins for the MAPK module and 20 proteins for PI3K module. For instance, loss of KRAS attenuated the activating phosphorylation of AKT, but not RSK or S6K (Fig. 6a).
- Independent parental KC clones differ little in terms of ERK or AKT activation (Ischenko et al., PNAS. 2014).
- The result with pAKT corroborates previously published data (Liao et al., JBC, 2006).
- Detecting p-ERK or p-AKT in RNA seq is not feasible. One can only make limited inferences regarding pathway activation.

9) Authors mention about p-YAP1 mediate p-ERK rescue. However, the experimental data is not provided.

- Yap1 data are now shown in Fig S1i.

10) Is figure 1I subcutaneous transplantation latency?

- Yes, a clarification has been made in Figure legend.

11) In figure 2B, please indicate how many different cell clones are transplanted to how many different mice. Again, is there only one single intact cell line which is transplanted into 10 different mice?

- This was done as requested (Fig. 2b). The transplanted cell lines are listed in Fig. S1e.

12) The metastasis data is quite interesting in figure 2D. A graph depicting % metastasis in mice which formed tumor would be better. Although the number of mice with tumor is low, it seems they develop metastasis with very high efficiency. How about tail vein injection of the KRAS intact/KO KPC cells into syngeneic mice? Would immune system still suppress primary foci formation in a Kras dependent manner in lung and Lymph nodes metastatic area.

- Fig. 2d has been changed as suggested.
- Loss of KRAS reduces the metastatic capacity of tumor cells by ~10 fold in nude mice and by >200 fold in syngeneic wild-type mice (Fig. S2a).

13) Figure S2B-C: Can iKras cells also form tumor in a different schedule. If so, would Doxycyclin withdrawal also impact metastatic foci? My overall question is how important is maintained Kras signaling also for metastatic foci? Can Kras therapies have an impact also on metastatic sites? The authors have great tools to answer these questions? Could they provide also an immune cell staining panel for the metastatic tumors of KRAS intact and WT mice?

- Immune staining of metastatic tumors has been added (Fig. S2j).
- Loss of iKRAS expression causes rapid extinction of metastatic tumors (new Fig. S2i).

14) The results with the MHC expression is quite intriguing. The researchers state that MHC dependent antigen presentation impact in immune response is arguable. However, recent publications indicate how important MHC presentation can be for both primary and metastatic tumors (e.g. Yamamoto et. Al. 2020, Autophagy promotes immune evasion of pancreatic cancer by degrading MHC-I, Pommier et al. Unresolved endoplasmic reticulum stress engenders immune-resistant, latent pancreatic cancer metastases. Science (2018)). From their RNAseq or RPPA results, did authors find any enriched pathways regarding the autophagy, ER stress mediated MHC expression regulation?

- We do not question the role of MHC genes in cancer immunity, quite to the contrary.
- The expression of autophagy pathway genes was not affected by KRAS loss ($p=0.2$ by two-tailed T test).

15) Figure 4/S4: Is KM signature correlated with both of the Ras dependency signatures? Authors only mention about the link in mutations observed in Kras and other inactivating tumor suppressor genes. Kras dependency difference in KM and KE tumors of TCGA is not nicely demonstrated in their analysis. Are Kras independent tumors KM tumors in figure 4a? Otherwise, just because SMAD4/2 expression is linked, can they state KM samples as Kras independent?

- Molecular and cellular differences between KRAS-dependent and KRAS-independent cancer cells have been extensively published (J. Settleman et al., R. DePinho et al., T. Jacks et al., F. McCormick et al.). We merely adopted the taxonomy.
- It is well recognized that the presence of epithelial markers in KE tumors is associated with KRAS dependency, while expression of mesenchymal markers in KM tumors is associated with KRAS independence, despite the presence of a KRAS mutation (J. Settleman et al., 2009; F. McCormick et al., 2018).
- As clearly stated on page 13, we designated KRAS-dependent tumors KE (epithelial) and KRAS-independent tumors KM (mesenchymal), as their histological examination confirmed their segregation into well, moderately and poorly differentiated (Fig. S4a).

16) Manuscripts subtyping pancreatic cancer based on transcription signatures indicate that tumors with EMT signatures (squamous, basal, quasimesenchymal etc...) have immune evasion phenotype or less immune cell infiltration. However, results here indicate an opposite output. Do they think a Kras dependency stratification of basal/squamous/quasimesenchymal type PDAC samples would distinctively divide immune cell phenotype? Does Kras dependency/activity results in a difference in Kaplan meier plot in TCGA samples??

- EMT and T cell infiltration are positively correlated across tumor types (Robinson et al., Nature, 2017; Wang et al., Nature Commun. 2018). However, much of the knowledge about PDAC subtypes (basal/QM), survival and the immune resistance needs to be reassessed (see Rev 3, point 2).
- Our data do not support a causative link between tumor morphology and immune infiltration, although both are related to KRAS loss. The key here is loss of KRAS

expression rather than EMT. Patients without the KRAS mutation survive significantly longer than those with mutant KRAS.

- In many studies infiltrating immune cells are grouped dichotomously as present or not. According to TCGA immune classification (The Immune Landscape of Cancer, 2018), KRAS-dependent tumors fall into four categories: C1 (termed wound healing), C2 (IFNG dominant), C3 (inflammatory) and C6 (TGF-beta dominant). In contrast, KRAS-independent tumors are significantly skewed towards the C3 subtype ($p=0.003$) (new Fig. 4f). This subtype is defined by a high Th1/Th2 ratio (in accordance with our data in Fig. 3), low to moderate tumor cell proliferation, and the most favorable prognosis (TCGA, Immunity, 2018).

17) Would antibody mediated CD8+ T cell depletion fully allow KRAS KO tumors to grow in syngeneic transplanted tumors?

- KRAS KO tumor rejection was studied in C57Bl/6 mice lacking CD4 or CD8 T cells. These mice fail to reject KRAS KO tumors (Fig. S2c).

18) What happens to other RAS isoforms when KRAS is knocked out?

- HRAS and NRAS are not affected at the RNA level or by protein expression (Fig. 1a). We performed RNA seq of ten cell lines and four tumors. Also, this matter has been investigated by T. Jacks (Nature Commun. 2017).

19) Total Ras activity assay is required in KRAS intact vs KO mice.

- This has been addressed in by T. Jacks and M. Pasca di Magliano. RAF/MEK/ERK activation can occur even in the absence of all three RAS genes (KRAS, HRAS, NRAS) (M. Barbacid. PNAS. 2014).

REVIEWERS' COMMENTS

Reviewer #1 (Remarks to the Author):

The revised manuscript addresses my previous concerns.

Reviewer #2 (Remarks to the Author):

The authors have addressed all my comments.

Reviewer #3 (Remarks to the Author):

In this revised manuscript, the authors have provided clarification on my comments and questions. I find this work of great interest given the multiple manipulations found. My comments were more focused on having the authors add more to the discussion about their findings and to better highlight their interesting work. I believe this work is of high caliber and should be published.

Specifically to their responses.

Major comments/questions

1. Given the multiple manipulations described, it might be useful to have an overall model in the final figure.

- This was done in Fig. S6.

I see the model here, but I think it would be more informative to show an integrated model in the main figures upfront or at the end as a main figure. It is a tremendous amount of work that has been shown and it would help the reader to better grasp the intricate pathways.

For example instead of just showing the individual KOs and phenotype it would be great to show specifically KRAS KO -> TGF-beta/EMT (Fig. 1 and 3) as an escape mechanism from KRAS ablation, but opens a new liability of immune sensitivity/detection (Fig. 2)

2. How do the authors reconcile EMT PDAC having a higher immune infiltrate (Fig 4), but in general we know these patients often have worsened survival?

Certainly the EMT spectrum of classical-basal is being defined and the papers (Chan-Seng-Yue et al. Nature genetics, 2020; Hayashi et al. Nature Cancer, 2020; Bhang et al., Nature Med. 2015) along with others (Porter RL et al. PNAS 2019; Ligorio M et al. Cell 2019) could be discussed and cited. My question is more that it is interesting that there is an escape from KRAS dependence that leads to higher immune infiltrates, but still we know that the vast majority of PDAC patients do not do well (irrespective of subtype), which as the authors note could be from TGF-beta mediated (below Q3)

3. Based on their analysis the higher immune cells in KM tumors have high PDL1 (CD274) and CTLA4 indicating that T-cells are suppressed by these two checkpoints. However, we know combined checkpoint in PDAC has not been effective. Do the authors have any potential ideas of why higher immune infiltrates do not correlate to better response to immunotherapy? Does this CD274 and CTLA4 elevation also occur in the mouse models of KRAS KO?

- TGF-beta has been identified as the principal barrier. Inhibition of TGF-beta by various means overcomes resistance to checkpoint therapy (e.g., Principe et al., Mol Cancer

Ther. 2019; Martin et al. Sci Transl Med. 2020).

- CD274 and CTLA4 elevation occurs in KRAS KO tumors by virtue of increased immune infiltration (Fig. S3c).

This is very interesting and should be expanded in the last paragraph of the discussion by describing an influx of immune cells, but with higher immune checkpoints that are able to maintain protection from the adaptive T-cell response. Potentially also highlight anti-TGF-beta PDAC clinical trials that are being done with immunotherapy.

4. Have the authors looked at metastatic potential of KRAS KO vs MUT cells via tail vein or orthotopic tumors? How about migration/invasion assays? Figure 1D would indicate that Tumor initiating capability is higher in KO. This finding would be very interesting and provocative since it would be counter to the dogma that targeting KRAS is universally a good idea...

- Both tail vein and orthotopic injections were performed (Fig. S1h and Fig. S2a). Loss of KRAS reduces the metastatic capacity of tumor cells (Fig. S2a).
- Figure 1d shows the opposite. Loss of KRAS reduces tumor initiating capability of cells.
- Targeting KRAS is a good idea. However, KRAS inhibition needs to be combined with immune activation.

Thank you for the clarification. Still interesting why KRAS KO have more LN mets. Or maybe this is just preference because KRAS MUT is able to seed lung at higher frequency?

Reviewer #4 (Remarks to the Author):

The authors have sufficiently responded to the raised concerns, included additional experiments and made important textual changes. This reviewer has no further suggestions.

Reviewer #1 (Remarks to the Author):

The revised manuscript addresses my previous concerns.

Reviewer #2 (Remarks to the Author):

The authors have addressed all my comments.

Reviewer #3 (Remarks to the Author):

In this revised manuscript, the authors have provided clarification on my comments and questions. I find this work of great interest given the multiple manipulations found. My comments were more focused on having the authors add more to the discussion about their findings and to better highlight their interesting work. I believe this work is of high caliber and should be published.

Specifically to their responses.

Major comments/questions

1. Given the multiple manipulations described, it might be useful to have an overall model in the final figure.

- This was done in Fig. S6.

I see the model here, but I think it would be more informative to show an integrated model in the main figures upfront or at the end as a main figure. It is a tremendous amount of work that has been shown and it would help the reader to better grasp the intricate pathways.

For example instead of just showing the individual KOs and phenotype it would be great to show specifically KRAS KO -> TGF-beta/EMT (Fig. 1 and 3) as an escape mechanism from KRAS ablation, but opens a new liability of immune sensitivity/detection (Fig. 2)

- This was done, Fig. 7.

2. How do the authors reconcile EMT PDAC having a higher immune infiltrate (Fig 4), but in general we know these patients often have worsened survival?

Certainly the EMT spectrum of classical-basal is being defined and the papers (Chan-Seng-Yue et al. Nature genetics, 2020; Hayashi et al. Nature Cancer, 2020; Bhang et al., Nature Med. 2015) along with others (Porter RL et al. PNAS 2019; Ligorio M et al. Cell 2019) could be discussed and cited. My question is more that it is interesting that there is an escape from KRAS dependence that leads to higher immune infiltrates, but still we know that the vast majority of PDAC patients do not do well (irrespective of subtype), which as the authors note could be from TGF-beta mediated (below Q3)

- We agree with the reviewer's point. There are two issues that affect the outcome: the amount of immune infiltrate and the level of immune activation. Our data show that KRAS ablation increases the influx of immune cells into the tumor, but with higher immune checkpoint expression. The exact role of TGF-beta in the suppression of adaptive T cell response will require further investigation.

3. Based on their analysis the higher immune cells in KM tumors have high PDL1 (CD274) and CTLA4 indicating that T-cells are suppressed by these two checkpoints. However, we know combined checkpoint in PDAC has not been effective. Do the authors have any potential ideas of why higher immune infiltrates do not correlate to better response to immunotherapy? Does this CD274 and CTLA4 elevation also occur in the mouse models of KRAS KO?

- TGF-beta has been identified as the principal barrier. Inhibition of TGF-beta by various means overcomes resistance to checkpoint therapy (e.g., Principe et al., Mol Cancer Ther. 2019; Martin et al. Sci Transl Med. 2020).
- CD274 and CTLA4 elevation occurs in KRAS KO tumors by virtue of increased immune infiltration (Fig. S3c).

This is very interesting and should be expanded in the last paragraph of the discussion by describing an influx of immune cells, but with higher immune checkpoints that are able to maintain protection from the adaptive T-cell response. Potentially also highlight anti-TGF-beta PDAC clinical trials that are being done with immunotherapy.

- This was done.

4. Have the authors looked at metastatic potential of KRAS KO vs MUT cells via tail vein or orthotopic tumors? How about migration/invasion assays? Figure 1D would indicate that Tumor

initiating capability is higher in KO. This finding would be very interesting and provocative since it would be counter to the dogma that targeting KRAS is universally a good idea...

- Both tail vein and orthotopic injections were performed (Fig. S1h and Fig. S2a). Loss of KRAS reduces the metastatic capacity of tumor cells (Fig. S2a).
- Figure 1d shows the opposite. Loss of KRAS reduces tumor initiating capability of cells.
- Targeting KRAS is a good idea. However, KRAS inhibition needs to be combined with immune activation.

Thank you for the clarification. Still interesting why KRAS KO have more LN mets. Or maybe this is just preference because KRAS MUT is able to seed lung at higher frequency?

- This matter is being investigated. The phenotype of KRAS KO cells may shed light on the requirement of KRAS for organ-specific metastasis.

Reviewer #4 (Remarks to the Author):

The authors have sufficiently responded to the raised concerns, included additional experiments and made important textual changes. This reviewer has no further suggestions.

Editorial Note: in our editorial assessment of the paper at the accept stage we noticed that there were some inconsistencies with the number of samples used for the analyses stated in the figure legends to Figure 4 and Supplementary Figure 6. We have asked clarifications to the Authors that have supplied a response (copied below), a Source data file and revised the manuscript accordingly. We have asked Reviewer #3 to comment on the authors' response and related changes.

Author's response:

The revised data relating to Figure 4 and Supplementary Figure 6 fully correspond to TCGA tumor or immune data and TCGA digital pathology data. This includes tumor purity, leukocyte counts, lymphocyte counts, and the immune subtypes. We have examined all tumors comprising the TCGA digital pathology database. New data have been added to Supplementary Figure 6 (6f and 6g) to strengthen conclusions. The results became clearer. Changes in the text are highlighted in red.

Our previous submission had an unfortunate problem, resulting in a convoluted presentation. The RDI>4 rule had not been exactly followed. It was a major oversight caused by the lack of proper attention. We now strictly follow this rule, even though the cutoff is arbitrary. No problem would have existed if we had set out a different score. We have examined TCGA data using two independent KRAS activity scores. These tumors relate to two overlapping datasets, which were generated using different techniques and various numbers of tumor samples: the digital pathology of tumor sections and the genomic analysis of immune cell infiltrates (Saltz et al., Cell Reports 2018; Thorsson et al., Immunity 2018). Accordingly, Figures 4d and 4e account for 110 tumors from the digital pathology dataset, while Figures 4f and 4g account for 120 tumors from the genomic dataset. Similarly, Supplementary Figures 6e, 6h and 6i account for 120 tumors from the genomic dataset. Considering the overlap between the two KRAS activity scores, we have examined >140 TCGA tumors. It is a significant number, and the results are transparent and straightforward.

The revision uses exact TCGA immune scores. This eliminates any ambiguity and sets standard for the analysis of secondary data. It is important to emphasize that the results are in line with the original submission, but they adhere exactly to the journal's requirements.

Reviewer #3 (Remarks to the Author):

I appreciate the model and overall the manuscript is written well.